



# Causes of uncertainties in the representation of the Arabian Sea oxygen minimum zone in CMIP5 models

Henrike Schmidt[1, 2], Julia Getzlaff[1], Ulrike Löptien[1, 2], and Andreas Oschlies[1, 2]

[1]GEOMAR Helmholtz Centre for Ocean Research Kiel, Düsternbrooker Weg 20, 24105 Kiel, Germany
[2]Kiel University, Christian-Albrechts-Platz 4, 24118 Kiel, Germany

**Correspondence:** H. Schmidt (hschmidt@geomar.de)

**Abstract.** Open ocean oxygen minimum zones (OMZs) occur in regions with high biological productivity and weak ventilation. They restrict marine habitats and alter biogeochemical cycles. Global models show generally a large model-data misfit with regard to oxygen. Reliable statements about their future development and the quantification of their interaction with climate change are currently not possible. One of the most intense OMZs is located in the Arabian Sea (AS). We give an overview
of the main model deficiencies with a detailed comparison of the historical state of ten climate models from the 5th coupled model intercomparison project (CMIP5) that present our present-day understanding of physical and biogeochemical processes. Considering a threshold of 60 $\mu$mol l$^{-1}$, we find a general underestimation of the OMZ volume in the AS compared to observations, that is caused by a too shallow layer of oxygen-poor water in the models. The deviation of oxygen values in the deep AS is the result of subduction of higher oxygenated waters in the Southern Ocean in the models compared to observations.
In addition, model deficiencies related to the coarse resolution of the abyssal ocean, are identified in the deep water mass transport from the Southern Ocean northward into the AS. Differences in simulated water mass properties and ventilation rates of Red Sea Water and Persian Gulf Water cause different mixing in the AS and thus influence the intensity of the OMZ. These differences also point towards variations in the parametrisations of the overflow from the marginal seas among the models. The results of this study are intended to foster future model improvements regarding the OMZ in the AS.

## 1 Introduction

Just like on land, marine animals also need oxygen to breathe, and they suffer if the oxygen concentration in the ocean falls below certain thresholds. Oxygen concentrations below the oceanic permanent thermocline depend on two mechanisms: (i) atmospheric oxygen enters the ocean at the surface mixed layer and is transported into the ocean interior by subduction and
mixing, and (ii) biological consumption by microbial respiration of sinking organic matter and respiration by higher trophic organisms. Main ventilation regions of the ocean are found at higher latitudes, where mode and deep water masses are formed (McCartney and Woodgate-Jones, 1991; Sverdrup, 1938). There is a close connection between age and oxygen concentration



of a water mass (Jenkins, 1977). The water mass age is defined by the time passed since the last surface contact, where its properties can be changed by gas exchange with the atmosphere. Older water masses typically feature lower oxygen concentrations because oxygen consumption has accumulated over longer time periods.

Worldwide there are three major regions with very low oxygen levels in the open ocean, so called oxygen minimum zones
(OMZs; e.g. Stramma et al., 2008). Those are located in the eastern tropical Pacific, eastern tropical Atlantic and the tropical Indian Ocean (IO). Typically, OMZs occur at intermediate depths between 100 and 1000 m where the respiration of exported organic matter is highest (Suess, 1980; Sverdrup, 1938). In the eastern tropical Atlantic and Pacific Ocean sluggish ventilation (Karstensen et al., 2008) and high biological consumption are drivers for the OMZs. The tropical IO differs from those ocean basins because it is bounded by the continent in the north and split into two basins by the Indian subcontinent. East of India,
the Bay of Bengal shows a shallow OMZ (∼200 to 600 m; Rao et al., 1994), whereas west of India, the Arabian Sea (AS) hosts one of the thickest OMZs in the global open ocean (∼200 to 1200 m). Compared to the other open ocean OMZs, the horizontal extent of the Arabian Sea oxygen minimum zone (ASOMZ) is relatively small. Nevertheless, it is considered to be one of the most intense OMZs due to its large vertical extent of oxygen-depleted water with oxygen concentrations typically around 3 $\mu$mol kg$^{-1}$ (Rao et al., 1994; Kamykowski and Zentara, 1990).

Various processes that determine the formation, maintenance and shape of the ASOMZ are already known from observations. The strong influence of the semi-annually changing monsoon winds on the circulation and resulting upwelling and subduction in the AS shapes the OMZ. A strong upwelling area is located off the Arabian peninsula and Somalia, which is associated with pronounced biogeochemical activity. A second upwelling region emerges along the south west coast of India during summer monsoon (Sharma, 1978; Shetye et al., 1990). During the winter monsoon downwelling occurs in the northern and northwest-
ern AS (Schott and McCreary, 2001; Hood et al., 2017). Also the surface circulation of the northern IO, which is well known from drifter data (Shenoi et al., 1999) and satellite altimetry (Beal et al., 2013), changes direction in response to the monsoon forcing (Schott and McCreary, 2001). The underlying subsurface ventilation pathways of water masses entering the AS are more uncertain due to a lack of observational data (McCreary et al., 2013; Schmidt et al., 2020).

The water of the ASOMZ comprises a variety of water masses with very different origins that are advected by the seasonally
changing current system (e.g. Hupe and Karstensen, 2000; You, 1997; Schmidt et al., 2020). Mixing analyses show that the bottom of the ASOMZ (below 1700 m) is predominantly ventilated by oxygen-rich Indian Ocean Deep Water (IODW, Acharya and Panigrahi, 2016). In the literature there are various definitions of IODW also referred to as Indian deep water. According to Schott and McCreary (2001) it is generated by deep upwelling of Circumpolar Deep Water and is a water mass that is specified for the northern IO. The Circumpolar Deep Water enters the Madagascar basin (Schott and McCreary, 2001) and according to
Tomczak and Godfrey (1994) IODW is transported northward along the western boundary, where it has water mass properties similar to North Atlantic Deep Water. Along the same route, just beneath IODW, Antarctic Bottom Water flows northward. In the northern hemisphere, IODW spreads eastward into the AS.

The dominant ventilating water masses influencing the upper ASOMZ are Red Sea Water (RSW), Persian Gulf Water (PGW) and Indian Central Water (ICW). The former two originate in the marginal Seas and enter the AS below the permanent thermo-
cline (Prasad et al., 2001; Beal et al., 2000; Shankar et al., 2005). They are easily defined by their respective salinity maxima.





ICW is subducted in the subtropics of the southern IO, spreads westward with the South Equatorial Current and is transported northward across the Equator with the Somali Current along the western boundary (Schott and McCreary, 2001) but also enters the AS from the east along the coast of India (Acharya and Panigrahi, 2016; Shenoy et al., 2020; Schmidt et al., 2020; Rixen and Ittekkot, 2005).

In addition to the physical parameters, the biogeochemistry is also subject to seasonality (e.g. primary production, Acharya and Panigrahi, 2016). Although many individual processes influencing the ASOMZ are already known, the interplay of these processes is still under discussion. What we do know, however, is that OMZs affect the ecosystem structure and reduce the habitat of higher trophic marine life (Levin et al., 2009; Stramma et al., 2012; Resplandy et al., 2012).

It is expected that global warming will intensify deoxygenation (Keeling et al., 2010; Bopp et al., 2013) and will also induce

changes in ventilation, stratification, and solubility. Eutrophication may drive enhanced microbial respiration, which in turn enhances deoxygenation (Breitburg et al., 2018; Keeling et al., 2010; Diaz and Rosenberg, 2008). The insufficient quantitative understanding of these processes results in uncertainties in the predictions of the extent and intensity of the OMZs.

For projections of further development of the OMZs and for the exploration of the interplay of different physical and biogeochemical mechanisms we rely on coupled biogeochemical ocean models. However, models seem to have a general problem in

estimating the oxygen content and changes in oxygen content in the ocean (Stramma et al., 2012; Séférian et al., 2020). The global deoxygenation trend, clearly visible in observations, as well as intensification and extension of OMZs with regional variations (Stramma et al., 2008, 2010; Keeling et al., 2010; Diaz and Rosenberg, 2008) is typically underestimated by Earth System Models (ESMs). In comparison to the observational trend the oxygen loss suggested by ESMs of the IPCC type is too weak and the simulated OMZ volumes differ substantially among models (Bopp et al., 2013; Cabré et al., 2015; Oschlies

et al., 2018, 2008). Especially for the IO there is no clear visible trend among a variety of models from the coupled model intercomparison project (CMIP; Oschlies et al., 2017), while global syntheses reveal a weak decrease of dissolved oxygen concentrations in the ASOMZ over the past decades (Ito et al., 2017; Schmidtko et al., 2017). There is some evidence that these model flaws are related to a deficient representation of ventilation pathways in models. On this basis, it is hardly possible to say whether the models' biogeochemistry does have deficiencies that are associated with the oxygen representation (Oschlies

et al., 2018; Segschneider and Bendtsen, 2013). If we look towards the future, the predictions regarding oxygen concentrations in the ocean differ considerably. Keeling et al. (2010) expect the global OMZ volume to expand, while for example Cocco et al. (2013) and Bopp et al. (2013) show that in many models, the volume of OMZs shrinks over the 21st century. With such large uncertainties, we cannot rely on future projections.

A first step to check the reliability of numerical models is to look how the models reproduce the current status. The presented

study identifies the major processes that are responsible for the uncertainties in the modelled oxygen with a specific focus on the physics in the IO. Therefore we assess the representation of the OMZ in the AS in the ten CMIP5 (coupled model intercomparison project phase 5) models that include a biogeochemical model component. These models summarize our present-day process understanding of the earth system and produce a fairly realistic large-scale picture of the global climate features. We aim to identify weaknesses of the ESMs that cause deficient oxygen concentrations. This will help to improve models and

future predictions, not only of the change of ocean oxygen concentrations. Specifically, this work focuses on the 3-D represen-



tation of the modelled OMZs and oxygen concentrations in the historical experiment of CMIP5. Furthermore, we classify the models systematically and identify similarities and differences in water mass representation and mixing among the models and with observations. We specifically target processes that are responsible for oxygen differences in the ASOMZ.

In section 2 we continue with a detailed description of the observational and model data considered, followed by the methods

in section 3. In the result-section 4 we compare the representation of the simulated ASOMZs in the CMIP5-models. Subsequently, we show the results and uncertainties of a water mass analysis in the core of the ASOMZ based on the observations. This analysis is then used to rate the model results, which were clustered to identify commonalities between models. The Discussion in section 5 puts these results into perspective to foregoing studies and to more recent CMIP6 model results and possibilities for further model improvements. We finish with summary and conclusions in section 6.

## 10   2   Data

### 2.1   CMIP5 simulations

The coupled model intercomparison project (CMIP5, Taylor et al., 2012) framework was designed to identify strengths and weaknesses of earth system models (ESMs) and thus improve climate predictions and identify uncertainties. Model output of dissolved oxygen was available from ten ESMs (Tab. 1) from the CMIP5 project (Taylor et al., 2012). The suite of model

simulations includes results from the Community ESM (CESM-BGC), two versions of the Geophysical Fluid Dynamics Laboratory ESM (GFDL-ESM2G/M), the Hadley Centre Global Environment Model (HadGEM2-ES), two versions of the Institute Pierre Simon Laplace ESM (IPSL-CM5A-LR/MR), two versions of the Max Planck Institute ESM (MPI-ESM-LR/MR), the Meteorological Research Institute ESM (MRI-ESM1) and the Norwegian ESM (NorESM1-ME). For References and further details see Tab. 1.

We focused on the so-called "historical" experiments, that were conducted for the years 1850 to 2005. From this time period we extracted the years 1900 to 1999 and consider the averaged model results for further analyses. This period is long enough for a robust calculation of the climatological mean state. Averaging also neglects the seasonal cycle. This is a reasonable approach for a uniform process analysis over the entire depth of the OMZ between 200 and 1800 m, as the seasonal cycle in oxygen concentrations is weak in the upper layers of the OMZ and not noticeable at greater depth. Next to dissolved oxygen,

temperature and salinity output from the same models was used in our analysis.

The CMIP5-ESMs differ in terms of the ocean circulation and biogeochemical modules. The horizontal resolution ranges from $2° \times 2°$ to $0.4° \times 0.4°$ and the number of resolved depth levels ranges from 31 to 63. Table 1 gives an overview of the circulation and biogeochemical model components and their resolution. In order to compare the model outputs with the observations, all model outputs were re-gridded to the same $1° \times 1°$ grid on which the observational data are interpolated (see below).





## 2.2 Observations

For comparison to the model results we use the global climatologies of dissolved oxygen, temperature and salinity climatologies provided by the World Ocean Atlas 2013 (WOA13). The climatological annual mean data cover a period from 1955-2012 and are available with a spatial resolution of 1° by 1° interpolated on 102 depth levels (Garcia et al., 2013; Locarnini et al., 2013; Zweng et al., 2013).

## 3 Methods

### 3.1 OMZ characteristics

As a first step, we compare the models and observations with respect to simulated oxygen in the AS. Depending on the process of interest, it is likely that different oxygen thresholds and the corresponding water volume need to be investigated. We thus compare the volume of the OMZ for a wide range of thresholds. We chose our threshold to be 50 $\mu$mol l$^{-1}$ to make it comparable to previous studies on CMIP5 oxygen distribution (e.g. Cabré et al., 2015; Cocco et al., 2013) and looked at the horizontal extension of the OMZ dependent on depth and the actual location of these areas in a map.

### 3.2 Cluster analysis

To reduce the high amount of data of the model output and detect similarities between the models and observations we performed a hierarchical cluster analysis. Here, the correlation between the vertical profiles of oxygen and salinity was used as the distance measure for the clusters. We are referring primarily to the curvature of the profiles and less to a systematic bias, e.g., an offset between profiles. For this purpose, the profiles are superimposed in such a way that the oxygen difference between the curves is minimal over the entire depth. This choice is motivated by the implicit assumption that the shape of the depth profiles contains more information on the underlying processes than the offset.

To determine the optimal number of clusters we used the silhouette-criterion (e.g., De Amorim and Hennig, 2015). The silhouette is a common measure of how closely a certain data point (here a profile) matches the data within its cluster and how loosely it matches the data in the other clusters. A large value close to one implies that a data point is in the appropriate cluster, while negative values indicate a wrong cluster choice. We calculated the averaged silhouettes for three to six clusters and selected the number of clusters with the highest average silhouette value. The resulting best choice of four clusters meets our visual rating. We performed the cluster analysis for oxygen profiles in the AS for all models and the observations. Furthermore, we used the same clustering method for the salinity profiles. Salinity is a conservative tracer that is useful when investigating mixing of water masses. Clustering of the models with respect to the modelled oxygen and salinity profiles helped to find similarities between the models and gave hints for typical model problems in this dynamically complicated region.

For this analysis we chose to exclude coastal areas and focus on the open ocean core of the ASOMZ in the central AS between 16 and 22 °N, 61 and 67 °E and from 10 to 1800 m depth and analysed averaged profiles in this region. To explain the dif-





ferences between the models, these analyses were complemented by water mass mixing analyses and an analysis of the water
mass properties in their formation region with respect to temperature, salinity and oxygen.

### 3.3 Determination of Water Masses in Models

Knowing the dominant water masses that mix in the ASOMZ, we analyse the representation of the respective water masses in
the individual models. Therefore, we localised the formation regions of the water masses in observations (Fig. 1 & supplement
Fig. S1-S3). Red Sea Water and Persian Gulf Water (RSW and PGW) are geographically restricted in their formation regions
(Fig. 1a) and thus easy to identify in the models. Indian Central Water (ICW) and Indian Ocean Deep Water (IODW) are much
less geographically restricted in their formation regions and it is likely that the formation regions in the models differ from the
formation regions that we know from observations. We used T-S diagrams of the whole IO to determine the T-S properties of
those water masses first from observations and applied the same procedure to the models (Fig. 2 & Fig. S4; see Tab. 2).
For the observations the respective values from the T-S diagram were compared to literature values (see Tab. 2). Figure 1
shows the grid boxes where these T-S properties are found in the IO in observations and thus give us the formation region.
Central waters are mixed water masses. We defined the upper and lower temperature and salinity limits of ICW using the linear
temperature and salinity relation that can be found in the T-S diagrams in the individual models and compared their formation
regions with the formation regions we obtained from the observations (Fig. 1b). For the calculation of the oxygen content of
ICW, we focused on the subduction area of ICW as prescribed in the literature (Schott and McCreary, 2001, Fig. 1b), to exclude
water parcels with similar T-S properties that are found in other areas in the IO. We also excluded the upper 200 m so that the
oxygen content of subducted ICW is not affected by the well ventilated mixed layer. Figure S2 shows the respective area for
the models and the deepest depth at each grid point, where the T-S properties are found.
IODW originates from the Southern Ocean, where it is often referred to as Circumpolar Deep Water, before it travels northward
in the deep IO. It is very cold and forms in great depth in the southern IO (Fig. 1c). Defining the T-S properties in the models
provides formation regions of IODW in the Southern Ocean below 1500 m depth, that differ slightly from those we find in
observations (Fig. S3). For the calculation of the oxygen content of IODW we include the local distribution differences of the
individual models. The oxygen content of the water masses is calculated, for each model, by the arithmetic mean of all grid
boxes of the corresponding source waters (see Tab. 2).

### 3.4 Analysing uncertainties of water mass mixing ratios

As we want to understand the physical mechanisms controlling the oxygen distribution in the different clusters, we looked at the
ventilation of the OMZ at different depths. Therefore, we carried out a water mass mixing analysis with the observations. This
serves to identify the ventilation depth of the individual water masses and their contribution. We used a linear mixing approach
and restricted the input to physical water mass properties from observational data. By considering potential temperature ($\theta$),
salinity ($S$) and mass conservation this yielded the possibility to resolve the mixing ratio of three different source water masses.





The set of linear equations was:

$$\theta = \alpha\theta_1 + \beta\theta_2 + \gamma\theta_3 \tag{1}$$

$$S = \alpha S_1 + \beta S_2 + \gamma S_3 \tag{2}$$

$$1 = \alpha + \beta + \gamma \tag{3}$$

$\alpha$, $\beta$ and $\gamma$ were the mixing ratio coefficients for each water mass. The equations were solved at each data grid point.

The three main source water masses in the AS are IODW, RSW and PGW and ICW (Fig. 2). There are two ways to determine the source water masses. First by taking values from the literature that are based on observations and second by taking the arithmetic mean of the WOA data in the IO as described in section 3.3 (Fig. 3, Table 2). For both sets of source water input properties the grid point data in the ASOMZ are the same data from the WOA13. We compare the results of both approaches

to obtain an uncertainty range of the water mass analysis that can be related to a change of the input. This allows us to draw conclusions on the sensitivity of the mixing in those models, where the source water properties deviate from the observations. The choice of the source water masses also restricts the resolvable water mass properties - it is not possible to mix the source water masses in a realistic way and get a higher/lower temperature and salinity than the highest/lowest temperature and salinity of the source water masses. With the described three source water masses this limits our analysis results to the central AS and

thus the core region of the ASOMZ, which is of the main interest of this study.

## 4   Results

### 4.1   Comparison of observed and predicted OMZs in the CMIP5 Models

For an overview of the differences in the oxygen distribution between models and observations, we calculated water volumes characterized by different oxygen thresholds in the AS westward of 79 °E (Fig. 4a). Eight out of 10 models underestimate the

volume of the OMZ and thus overestimate the oxygen content of the water, especially for oxygen thresholds above 50 $\mu$mol l$^{-1}$.

The vertical extent of low oxygen waters characterized by the 50 $\mu$mol l$^{-1}$ threshold is compared by area-depth profiles in the AS (Fig. 4b). Observations show oxygen values below 50 $\mu$mol l$^{-1}$ in the depth range between 200 and 1800 m (Fig. 4b). The maximal extent of the OMZ is around 900 m. Below 900 m the oxygen content is overestimated compared to the observations

in nearly all models. Above 900 m the models split up in two groups, where one group overestimates the extent of the OMZ and the other underestimates it. To investigate this model-data misfit further we focus on the depth horizon of the OMZs in the models and look at the horizontal expansion of the OMZs there. Figure 4c shows the spatial OMZ extend at 500 m depth, where the model-data misfit is the largest. Four out of ten models (IPSL-CM5A-MR,IPSL-CM5A-LR, HadGEM2-CC, MRI-ESM1) generally simulate so high oyxgen values over the whole water column that there is no water with oxygen concentrations less

than 50 $\mu$mol l$^{-1}$ in 500 m depth (Fig. 4c). The models that overestimate the OMZ area of less than 50 $\mu$mol l$^{-1}$ show too low oxygen values compared to observations in the whole AS and a southward expansion of the OMZ with one exception:





In the NorESM1-ME model the OMZ is shifted to the south-eastern boundary of the AS and is located between 15° N and the equator (Fig. 4c). All in all this wider horizontal expansion of the oxygen-poor areas below 50 $\mu$mol l$^{-1}$ in the models compared to the observations (Fig. 4c) cannot compensate for the reduced thickness of the oxygen-depleted layers, which is responsible for the overall underestimated OMZ volume in the AS in the CMIP5 models (Fig. 4a).

Thus the oxygen distribution differs considerably among the CMIP5 models in the AS. None of the CMIP5 models reproduces the observed oxygen distribution. Also the extent of the OMZ depends highly on the chosen threshold (Fig. 4a). For a more general comparison of the models with each other and with the observations, we therefore decided to use averaged oxygen profiles in the AS for the cluster analysis.

## 4.2 Cluster analysis

We performed a cluster analysis to identify commonalities between the models. Figure 5 shows these profiles averaged in the box in the core region of the ASOMZ as shown in Fig. 4c. Based on the silhouette-criterion (see section 3.2) we obtain four clusters. The naming of the clusters is based on their agreement with the observations. Cluster *HIGH* and *MEDIUM* are the largest clusters. Cluster *HIGH* groups with the observations and contains the CESM1-BGC, GFDL-ESM2G and MPI-ESM-MR/LR. Cluster *MEDIUM* contains the HadGEM2-CC, GFDL-ESM2M and IPSL-CM5A-MR/LR. In addition, two outliers

were identified, that each form their own cluster: MRI-ESM1 (cluster *LOW1*) and NorESM1-ME (cluster *LOW2*).

The surface oxygen concentration of the models is similar among the models in the AS but about 25 $\mu$mol l$^{-1}$ higher than in observations (Fig. 5a). Below 1800 m all models (except IPSL-CM5A-MR) show too high oxygen concentrations compared to observations (Fig. 5a).

The main difference between the clusters is noticeable in the oxygen content from around 250 to 1300 m depth in the core

of the OMZ. The observations show oxygen concentrations close to zero over these depths. Cluster *HIGH* models also have an averaged oxygen concentration close to zero, but not all cover the full depth of the observational core. Cluster *MEDIUM* models generally show higher averaged oxygen concentrations above 80 $\mu$mol l$^{-1}$ in three out of four models in comparison to cluster *HIGH*. The model from cluster *LOW1* has even higher oxygen concentrations and the model in cluster *LOW2* has an averaged oxygen minimum that is found in shallow depth around 400 m (Fig. 5a).

To differentiate between physical and biogeochemical processes, we also carried out the cluster analysis for the salinity profiles. The cluster analysis for the salinity profiles (Fig. 5b) groups nearly all models and the observations into the same clusters as the cluster analysis for oxygen. Only the GFDL-ESM2G changes from oxygen cluster *HIGH* to salinity cluster *MEDIUM*. In contrast to the averaged oxygen profiles the simulated averaged salinity profiles (Fig. 5b) are close to observations below 1800 m. Between 800 and 1800 m nine out of ten models underestimate the salinity. Above 800 m the model data show differ-

ent patterns in over-/underestimating the salinity in the AS, which differentiate the individual clusters. Three out of four cluster *HIGH* models overestimate the salinity up to the upper boundary of the OMZ (Fig. 5b). In contrast to that all cluster *MEDIUM* models overestimate the averaged salinity at depths around 400 m in the AS. Cluster *LOW1* has even higher salinity values than the models from cluster *MEDIUM*. The model from cluster *LOW2* underestimates the salinity all the way up to the surface.

The clustering of the models reveals a connection between the representation of oxygen and salinity in the CMIP5 models





with one exception (GFDL-ESM2G). The grouping of the models from cluster *HIGH* with the observations indicates that the circulation in this group is similar to the real circulation, or at least that we could not identify any fundamental problems in the circulation models. Still the OMZs of the models from cluster *HIGH* differ in shape and extension compared to the observational OMZ. In contrast, the cluster analysis indicates deficiencies in the circulation model that are responsible for deficiencies

in simulated oxygen, for the models in clusters *MEDIUM*, *LOW1*, and *LOW2*, that do not group with the observations regarding salinity. However, this does not exclude the possibility that the biogeochemical model components of these models also have problems. These are just not clearly identifiable due to the underlying uncertainties in the physical model components. To gain further insights in the water masses and mixing processes we analysed the water mass representation in the models.

### 4.3 Water Mass Representation in Models

We concentrate on the three main water masses that mix in the ASOMZ which are IODW, ICW and RSW and PGW. The water mass mixing analysis (Fig. 3) shows that IODW is the dominating water mass in the deep AS and ICW and RSW and PGW have an increasing influence on the OMZ with decreasing depth. The percentages of the water masses with depth and their uncertainties are explained in more detail in section 4.4.

IODW forms in the Southern Ocean, where it is often referred to as Circumpolar Deep Water. Its temperature varies from 0 to

1 °C and its salinity from 34.65 to 34.7 (Table 2). All models reproduced these characteristics fairly well. Also the formation region (Fig. 1c & Fig. S3) is correctly simulated by all models except NorESM1-ME, where these properties do not reach deep enough in the southern IO but a large amount of water with these characteristics is found in the equatorial eastern IO.

The simulated oxygen concentrations of IODW vary between 181 (IPSL-CM5A-MR) and 301 $\mu$mol l$^{-1}$ (NorESM1-ME; Table 2, Fig. 6). The observational mean oxygen concentration is 200 $\mu$mol l$^{-1}$ (Table 2, Fig. 6). Figure 6 shows a comparison of the

oxygen concentrations at the bottom of the ASOMZ at 1800 m depth and the oxygen concentrations of IODW in its formation region. The offset between those two concentrations indicate that the respiration of organic matter during the transit from the formation region of IODW to the central AS results in an oxygen consumption of 136 $\mu$mol l$^{-1}$ in the observations. In cluster *HIGH*, all models show offsets in the oxygen concentration between IODW and the bottom of the ASOMZ that are similar to the one found in the observations. For the majority of cluster *MEDIUM* models (IPSL-CM5A-MR/LR, HadGEM2-CC) and

cluster *LOW2* the offset in the oxygen concentration is smaller compared to the one in observations. In cluster *HIGH* we are not sure why the simulated oxygen is not closer to the observations. Even if the salinity profiles cluster with the observations and the offset in oxygen between the formation region of IODW and the deep AS resembles the observations, we cannot exclude deficiencies in the physics, especially in the abyssal ocean. However, what is immediately noticeable in Fig. 6, is that IODW almost systematically has an offset in the Southern Ocean.

To find out more about the differences between clusters in the oxygen consumption of IODW on the way to the AS, we looked at the age since surface contact of two models. The GFDL-ESM2G (cluster *HIGH*) has an average age of 101 yrs of water in the formation region of IODW in the deep Southern Ocean and an average age of 579 yrs of the deep water in the AS. In the GFDL-ESM2M (cluster *MEDIUM*) the respective water mass ages are older with 252 yrs and 780 yrs, respectively. The age difference between the formation region and the AS are 478 yrs (GFDL-ESM2G) and 528 yrs (GFDL-ESM2M). This





shows that the two models start with differently aged water, which already might explain the lower oxygen concentration of GFDL-ESM2M in the Southern Ocean. Both models have the same biogeochemical model component and the same horizontal resolution of the physical model component, but the GFDL-ESM2G has a higher vertical resolution. This also suggest, that the vertical resolution has an impact on the water mass formation in the Southern Ocean and suggests that the circulation

differs among the two models and thus also the residence time and possibly also the export production. For clusters *MEDIUM*, *LOW1*, and *LOW2*, we have already obtained many indications that there seem to be deficiencies in the ventilation and water mass mixing of the ASOMZ. In clusters *MEDIUM* and *LOW2* these offsets in the Southern Ocean seems to be additionally superimposed by uncertainties in oxygen consumption in the abyssal ocean.

RSW and PGW are straightforward to define in models, as they have a distinct origin in the Red Sea and the Persian Gulf,

respectively (Fig. 1a & Fig. S1). The temperature range between 18 and 30 °C in observations is well represented in all models (Table 2). However, the simulated salinity in the formation region varies among the models. While the lower limit of 37.14 in observations is met by most models (Table 2), the upper limit varies from 39.28 (MPI-ESM-LR) to 46.71 (IPSL-CM5A-MR). In general, we find an overestimation of the salinity of RSW and PGW in all clusters.

One consequence of more saline and thus denser water is that it might ventilate the OMZ at a different depth or generate

salinity maxima that are not found in observations. The averaged salinity profiles in the AS confirm this overestimation of salinity especially for cluster *MEDIUM (LOW1)* in depth between 200 - 500 (1000) m depth (Fig. 5b). For cluster *LOW1* the deep reaching salinity overestimation cannot be explained by offsets in the source water mass properties alone, although the peak at around 500 m depth coincides with the depth of maximal water mass contribution of RSW and PGW (Fig. 5b). A possible further explanation would be enhanced mixing of RSW and PGW into the AS and also stronger evaporation/less

precipitation over the AS. Below 500 m, the reduced salinity and the mixing analysis indicate less input of RSW and PGW in nearly all models compared to observations. This deficit would therefore have to be compensated by another water mass that is mixed into the ASOMZ.

The mean oxygen content of RSW and PGW is quite similar among the models but has a considerable positive offset compared to observations of up to 87 $\mu$mol/l (Table 2). While the observations show a mean oxygen content of 128 $\mu$mol/l the models

have a mean oxygen concentration between 179 (CESM) and 215 $\mu$mol/l (NorESM1-ME). The concentration differences between the clusters are comparable to those within the clusters, even though the models in cluster *HIGH* tend to have lower oxygen concentrations than those in cluster *MEDIUM*. This higher oxygen concentration in the formation region combined with the modified mixing of water in the ASOMZ due to density changes by overestimated salinities of RSW and PGW in cluster *MEDIUM* serves to explain the enhanced oxygen in the ASOMZ (Fig. 5a). In clusters *LOW1* and *LOW2*, RSW and

PGW have the highest oxygen concentration of all models.

ICW is subducted in the southeastern IO in the subtropical cell region (Fig. 1b & Fig. S2). Central water masses can be recognised by their linear T-S relationship. Table 2 gives the upper and lower temperature and salinity limits of ICW for each model. The temperature range we find in observations (7.7 - 15.8°C) is 2.2 degree below the established literature value. The temperature range of the models thus corresponds to those values going from 7 °C (IPSL-CM5A-LR) to 19.9 °C (NorESM1-

ME). Also the salinity corresponds to a great extent with values from 34.57 to 34.57 in observations and 34.49 (GFDL-ESM2G)





to 36.13 (CESM-BGC) in models. For both properties the clusters show no clear separation among each other (Tab. 2). The mean oxygen content of ICW of the models spreads from 170 $\mu$mol l$^{-1}$ (CESM-BGC) to 233 $\mu$mol/l (HadGEM2-CC) which brackets the observational concentration of 200 $\mu$mol/l. Again, no clear separation between the clusters is noticeable.

### 4.4 Uncertainties of water mass mixing ratios impacting the OMZ according to observations

We performed the water mass analysis for the observations using two different sets of the source water mass properties as input. The first input comes from established literature values (see Tab. 2 for input and Reference; Fig. 3a). The second input is derived from WOA13 data (Fig. 3c; section 3.4). This enables us to estimate the sensitivity of the analysis to differences in assumed water mass characteristics.

Starting with the input from literature values, the impact of IODW on the lower ASOMZ is dominating with a water mass
contribution of up to 80 % (Fig. 3b). IODW has still an impact of about 50 % at intermediate depths below 800 m, but is barely found at the upper boundary of the ASOMZ at depths of 200 m, where the intermediate water masses such as ICW and RSW/PGW are dominating (Fig. 3b). This holds especially for ICW, which has a maximal contribution at the upper boundary of the ASOMZ of about 80 % and decreases downward to a fraction of less than 20 % at 1800 m depth. Above 500 m depth RSW and PGW contributes between 15 and 40 % to the mixed water in the OMZ (Fig. 3b). This fraction is decreasing with
depth tending towards 0 % at the bottom of the ASOMZ.

The spatial variability of the composition of water masses is more variable in the upper layers of the ASOMZ. This is due to the fact that temperature and salinity in the deep ocean vary less than in the thermocline and permanent thermocline, which are affected by heat and freshwater fluxes, seasonal variations and turbulent mixing.

Switching to the source water mass definitions defined from the WOA data (Fig. 3c; section 3.4) the greatest deviation of the
input parameters is for RSW and PGW (Fig. 3b,d). This water has mean temperature and salinity values of 24.1 °C and 38.9, which is higher by 5.4 °C and 2.2 compared to the literature values of Hupe and Karstensen (2000). For ICW the temperature value defined from the mean WOA data is 15.8 °C and thus lower than the literature value of Acharya and Panigrahi (2016).

Despite the change in source water definitions, the impact of IODW on the ASOMZ in the second analysis is similar to that in the first analysis indicating a robust result for the mixing ratios diagnosed from observations (Fig. 3d). The impact of ICW on
mixing ratios in the ASOMZ is generally a few percent higher throughout the water column in the second analysis compared to the first one. The largest differences however are noticeable for RSW and PGW at depths between 200 and 600 m, where the maximal contribution is 20 % with WOA13 input. This is just half as much RSW and PGW compared to the literature values that mixes into the ASOMZ.

Comparing the outcome of these two water mass analyses gives a stable result for the mixing of water masses in the deep AS.
It is more sensitive to varying source water mass characteristics at intermediate depth and in particular, to fluctuations in water masses whose properties differ significantly from those of the other water masses being mixed. As seen in section 4.3 RSW and PGW is by far the saltiest and warmest water mass but also the T-S properties of this water mass differ most clearly among the models. This can result in uncertainties in the mixing ratio of the water masses in the models in the ASOMZ. Since the water





masses are of different origins and also have different oxygen concentrations, this can affect the simulated oxygen content of the OMZ.

## 5  Discussion

Previous global and regional studies point out that CMIP5 models systematically overestimate the volume of OMZs (e.g. Bopp
et al., 2013 (global OMZs); Cabré et al., 2015 (Pacific OMZs); threshold of 50 $\mu$mol l$^{-1}$). We cannot support the statement with our findings for the AS. All ten models underestimate the ASOMZ volume when we consider oxygen thresholds of 60 $\mu$mol l$^{-1}$ or higher (Fig. 4a). In contrast, Rixen et al. (2020) and our own analysis here show that two models (CESM1-BGC, MPI-ESM-LR) overestimate the OMZ volume when considering hypoxic conditions (<20 $\mu$mol l$^{-1}$), with the maximum simulated OMZ volume being more than twice as large as in observations. However, the other eight models still underestimate the OMZ
volume for the lower threshold. This is consistent with our analysis of the same 10 models (Fig. 4a). We find that this volume underestimation in the models is mainly caused by an OMZ that has too small a vertical extent (Fig. 4b). Previous studies (e.g. Kamykowski and Zentara, 1990; Rao et al., 1994) that included observations pointed out that the core of the ASOMZ was thicker than in the Atlantic and Pacific OMZs with oxygen values below 5 $\mu$mol l$^{-1}$ expanding over a depth range of about 1000 m. This large vertical expansion causes the horizontally confined ASOMZ to have such a large volume. However, only
one in ten models is able to completely cover this depth of oxygen depleted water (MPI-ESM-LR; Fig. 5a).

Recent studies analysing CMIP5 and CMIP6 model data show that increasing the horizontal resolution does not overcome the major problems with respect to simulating oxygen in the open ocean. Despite better representation of mesoscale processes due to higher resolution, the expected improvement in oxygen representation is absent in the CMIP6 models on a global scale (Séférian et al., 2020). While the model-data misfit for the upper ocean oxygen content was reduced from the CMIP5 to CMIP6
model versions in the Indian and Pacific Ocean, Séférian et al. (2020) suspects a systematic bias in biogeochemical models due to sign shifts in model-data deviations between the two CMIP phases in the Atlantic Ocean, where the CMIP5 models simulated a stronger-than-observed OMZ and the CMIP6 models a weaker-than-observed OMZ. Among the models considered here, we confirm the lack of an apparent correlation between model resolution and better representation of the OMZ in the IO, because we cannot establish a relationship between the oxygen clusters and the respective resolution of the models (Tab.
1 & 2). However, we must take into account that all CMIP5 models are far from eddy resolving and inclusion of mesoscale processes in the CMIP6 models brought only moderate improvements in subsurface oxygen representation (Kwiatkowski et al., 2020).

To simulate the OMZ accurately, both the physical (ventilation) and biogeochemical components (respiration) must be adequately represented in the models. Starting with the water masses that contribute to the ASOMZ, errors in deep water mass
formation and transport can result in an incorrect representation of the OMZ. A major CMIP5 model problem that we could identify and link to ASOMZ is the higher-than-observed oxygen content in the Southern Ocean, which is reflected in the deep AS. We find this tendency in all models and there is no cluster dependency (Fig. 6). Kwiatkowski et al. (2020) and Tagklis et al. (2020) state that the spin-up times of CMIP5 models are not long enough to equilibrate biogeochemical conditions in





the deep ocean. Mignot et al. (2013) shows that physical properties and the large-scale circulation are already in equilibrium after 250 yrs, whereas Séférian et al. (2016) shows that this does not hold for biogeochemical tracers. Moreover, the drift is highly model dependent and not directly correlated to the spin-up times that range from 500 (HadGEM2-CC) to 11900 yrs (MPI-ESM-LR). Further uncertainties are linked to a generally coarse vertical resolution of the deep ocean, that shape the

bottom topography and limit biogeochemical processes related to the bentho-pelagic ecosystem (Kwiatkowski et al., 2020). The coarse resolution can influence the export pathways and thus timescales of IODW from the Southern Ocean northward into the AS and the bentho-pelagic ecosystem defines the oxygen consumption rate on its way and causes oxygen concentration differences in the deep AS. In our study we also cannot find a connection between the model spin-up times and the oxygen change during the 20th century in the AS and the OMZ representation in the historical experiment of the models, especially

not in the deep AS (Fig. S5). Nevertheless, there are opposing oxygen trends also in the deep AS in all models between 1900 and 1999 but they are small compared to the trends in the thermocline and the OMZ layer (Fig. S5). The oxygen differences between the formation region of IODW in the Southern Ocean and the bottom of the ASOMZ (Fig. 6) are close to the observed oxygen differences for the clusters *HIGH* and *LOW1*, but smaller for the clusters *MEDIUM* and *LOW2*. The comparison of the two GFDL-ESMs (cluster *HIGH & MEDIUM*), which have the same biogeochemical model component, shows similar

oxygen offsets but different oxygen concentrations in the Southern Ocean (Fig. 6) and also a difference in water mass ages in the Southern Ocean of 150 yrs. The age difference between the two models in the deep AS is only 50 yrs. This suggests that the circulation differs in both models and thus also the residence time what would influence the consumption rate on the way northward from the Southern Ocean. We therefore deduce that in clusters *MEDIUM* and *LOW2*, circulation is responsible for a large part of the oxygen differences in the deep ASOMZ.

Besides transport times and pathways of water masses into the ASOMZ, we find uncertainties in the formation of water masses among the models. RSW and PGW oxygen concentrations were quite far off in the formation regions. The observations show a strong decrease of oxygen from around 200 $\mu$mol l$^{-1}$ at the surface down to 50 $\mu$mol l$^{-1}$ in 300 m depth. This oxygen decrease is only captured by two models (CESM1-BGC, GFDL-ESM2G). In the other eight models, the oxygen concentration is uniformly distributed throughout the water column.

Furthermore, coarse resolution models generally prescribe the overflow through small channels that are not resolved by the grid resolution. This is also the case for the outflow of RSW and PGW. Seland et al. (2020) find a too warm and saline core in the AS in subsurface depth in the CMIP6 version of the NorESM and trace it back to the outflow of the Red Sea. They state that such subsurface ocean biases can be linked to the coarse ocean resolution and deficiencies in process parametrisation. We can find similar patterns in our study with a too saline layer in cluster *MEDIUM* and *LOW1* above 500 m depth (Fig. 5), which

is likely caused by the inflow of RSW and PGW. This points towards a problem in the parameterisation of the outflow of RSW and PGW at least in the clusters *MEDIUM* and *LOW1*. In addition, the higher-than-observed salinity could be strengthened by the positive salinity offset in models compared to observations in the source regions of RSW and PGW, which we found in all clusters. Somewhat surprisingly, 8 of 10 models from all clusters show less saline water than the observations in the layer between 500 and 1800 m depth (Fig. 5), which might be explained by enhanced ventilation with other water masses such as

ICW.





The other intermediate water mass that ventilates the ASOMZ is ICW, which is subducted in the subtropical cell region in the south eastern IO. Propagating westward and northward into the AS it likely mixes with other intermediate water masses in the subtropical and tropical IO. Sallée et al. (2013a & 2013b) examine the circulation and water mass formation in CMIP5 models in the Southern Ocean. They found a warm bias in the subtropical region in nearly all models and a too strong seasonal

cycle of the subtropical mixed layer. According to Sallée et al. (2013b) this causes excess subduction of too light mode water in the western basin, that gets denser in the eastern part of the basin. Further, the total amount of subtropical water in the models is underestimated. The models considered here show no sign of subduction of too light ICW, because the water mass characteristics within the area where ICW is subducted in all models from all clusters fit to the observations. However, this is just one water mass that is formed and transported in the IO subtropical gyre. To give clear statements about the mixing and

its properties when ICW reaches the ASOMZ, further investigations on the subtropical and tropical IO water wasses in CMIP5 models would be necessary.

More uncertainties can be found in the water mass formation in the Southern Ocean in CMIP5 models, where we located the source of IODW. For the best possible comparison with other studies on the CMIP5 models, it is meaningful to use and discuss not only circumpolar deep water (CDW) but also Antarctic Bottom Water (AABW) as a source for IODW. First, this

is reasonable because the water mass properties of CDW (1.85 °C, 34.69; multi model mean from Sallée et al., 2013b) and AABW (0.18 °C, 34.72) overlap with our and the literature's definition of IODW. Second, the term IODW is often only used in the AS and CDW and AABW both flow along the western margin towards the north and could thus mix on the way to become IODW.

Sallée et al. (2013b) find large variations among CMIP5 models for CDW and bottom water in the Southern Ocean. With one

exception (HadGEM2-CC) all models underestimate the volume of CDW with a multi model mean volume ($25.2 * 10^{16} m^3$) that is about 77% of the observed volume. If we look at the CDW volume of the individual models considered in our study, we notice no clear differences in volume between the individual clusters. Most of the models have a volume of CDW that is just below the multi model mean value of Sallée et al. (2013b). However, it is noticeable that the GFDL-ESM2M has a surprisingly small volume of CDW ($\sim 1.6 * 10^{16} m^3$), which is probably balanced by a larger amount of intermediate water.

This imbalance would result in changes in circulation, reinforcing our conclusion that uncertainties in oxygen are primarily caused by circulation in the models in cluster *MEDIUM*. For AABW, that is transported northward along the western boundary together with CDW, Sallée et al. (2013b) find larger variations in its volume of the individual models than for CDW and also the multi model mean volume exceeds the one estimated from observations ($5.5 * 10^{16} m^3$). Two models, one from cluster *HIGH* (GFDL-ESM2G ) and the model from cluster *LOW2* (NorESM1-ME) overestimate the volume of AABW by far

($\sim 14 * 10^{16} m^3$). Sallée et al. (2013b) find also differences in the location of the bottom water with a bottom layer that rises to the surface (IPSL-CM5A-LR, IPSL-CM5A-MR) against a deep thin layer of concentrated bottom water in high latitudes (HadGEM2-CC). They state that these huge differences are linked to different parametrisations of convection and formation of deep waters and that, so far, no ESM has been able to correctly simulate the properties of the abyssal oceans. Looking at their results and sorting them into our cluster analysis we identify no clear differences in volume of AABW between the individual

clusters of our study (e.g. volume of AABW in MPI-ESM-LR (cluster *HIGH*) and in HadGEM2-CC (cluster *MEDIUM*) is





nearly similar with $\sim 6 * 10^{16} m^3$). What we see in Fig. 6 are too high oxygen values in the southern IO in the area that we assumed for IODW formation. This might be linked to the excessive amount of AABW, which should be generally younger than CDW, because it is recently ventilated in the Southern Ocean or at least should be according to observations, and thus contains more oxygen. For the models from clusters *HIGH* and *LOW2* we see this positive oxygen offset propagating into the

deep AS (Fig. 6). However, for the models from the other two clusters the uncertainties in bottom and deep water do not show a direct link to the oxygen concentration in the Southern Ocean, as we were not able to identify a cluster-dependent oxygen difference there but rather clear differences in the consumption rate on the way to the OMZ (Fig. 6). Thus the Southern Ocean water mass circulation might influence the sequestration and transport of heat, salt and nutrients into deeper layers and change the mixing and feedback cycles in the interior ocean. A more detailed look at the parametrisation of the Southern Ocean in

ESMs and the connected biogeochemical feedback cycles is beyond the scope of this paper but should be address in a future study.

What has not yet been taken into account in this analysis and might influence the supply of oxygen from below to the OMZ are additional possible deficiencies in upwelling in the AS. Too strong upwelling of oxygenated deep and bottom water from below would flatten and weaken the ASOMZ. You (2000) and Stramma et al. (2002) find a deep overturning circulation in the

AS with inflow below 2500 m depth and an overlying outflow between 300 to 2500 m depth. Stramma et al. (2002) state that the rising bottom water in the AS reduces its oxygen content by mixing with the less oxygenated intermediate waters. However, they point out that there are large uncertainties associated with computing the strength of the overturning cell. Thus there is no reference value for upwelling strength in the AS we could compare with the CMIP5 models. This would need further investigation from the observational perspective.

Another point that has not been examined in detail here, but which emerges from the analysis, is an overestimation of biogeo-chemical oxygen consumption in the AS in the models in cluster *HIGH*. In contrast to the models from clusters *MEDIUM*, *LOW1*, and *LOW2* where the analysed physical processes can explain much of the model-data misfits in oxygen concentrations, we find no obvious errors in the physical processes in the cluster *HIGH* models. They nevertheless show a greater-than-observed oxygen decrease in the lower oxycline at the bottom of the OMZ (Fig. 5a) what can be caused by excessive oxygen consump-

tion. For clusters *MEDIUM*, *LOW1*, and *LOW2* we cannot make any inferences about the interaction of the biogeochemical model component with the uncertainties in the physical model component.

## 6 Summary & Conclusions

In this paper we compared 10 ESMs from the CMIP5 historical experiment and analysed the 3-D representation of the modelled OMZs in the AS. We sorted the models systematically with a cluster analysis and identified similarities and differences in water

mass representation and mixing among the models and with observations. We identified weaknesses of the ESMs that cause deficient oxygen concentrations in the AS in the northern IO and looked for similarities among the models. We found that, in particular, excessive salinity in the Persian Gulf and the Red Sea in the models leads to different water mass mixing in the ASOMZ than in the observations. In addition, the overestimated oxygen content in the Southern Ocean leads to the OMZ being





fed with more oxygenated water from below in the models.

We found large uncertainties in the oxygen representation in the AS among the CMIP5 simulations. Overall the underestimation of the OMZ volume is generally caused by a simulated OMZ that is too shallow compared to observations.

We further analysed the source water mass properties in the marginal seas, the southern IO and in the subduction region of
ICW. While several models show obvious deficiencies in reproducing circulation patterns, the water mass transport into the AS and the mixing due to density uncertainties in the source water masses, these deficiencies on their own are insufficient to explain the deviating oxygen concentrations. Our results also deduce overconsumption of oxygen in the biogeochemical model components in the AS, especially where the physical model components show no obvious deficiencies in circulation and mixing, while the oxygen concentrations deviate in the AS. Since the next generation of CMIP models, that has higher
resolution, tends to overestimate oxygen concentrations in the AS as well, our analysis points out other processes in addition to the consideration of mesoscale features need improvement for a better representation of the ASOMZ.

We conclude that model-data misfits in oxygen can be caused by errors in the physical models, which are summarised in Fig. 7. These include the circulation and water mass formation in the Southern Ocean, the deep water mass transport and resolution of the abyssal ocean and parametrisation of overflow in narrow straits. We consider it useful to first address local processes
that can be clearly delimited and whose uncertainties are not amplified by other errors. These are the parametrisation of the overflow of RSW and PGW and their T-S properties in the source region. We hope that this process improvement can reduce the model-data misfit and diminish the uncertainties in future oxygen projections.

*Code and data availability.* The CMIP5 model output is publicly available at https://esgf-node.llnl.gov/projects/cmip5/. The WOA13 data are available at https://www.nodc.noaa.gov/OC5/woa13/woa13data.html. The code is available at https://oceanrep.geomar.de/52412/

*Author contributions.* H. Schmidt, J. Getzlaff, U. Löptien, and A. Oschlies conceived the study. H. Schmidt handled all the data and performed the calculations. All authors discussed, wrote and modified the manuscript.

*Competing interests.* The authors declare that they have no conflict of interest.

*Acknowledgements.* We acknowledge the World Climate Research Program's Working Group on Coupled Modelling, which is responsible for CMIP, and we thank the climate modeling groups (see section 2.1) for producing and making available their model output. For CMIP the
U.S. Department of Energy's Program for Climate Model Diagnosis and Intercomparison provides coordinating support and led development of software infrastructure in partnership with the Global Organization for Earth System Science Portals. We would like to thank Heiner Dietze for his support in data acquisition and processing, and for the helpful discussions. We would like to thank Nicole Köstner for her help with the



final editing of the graphics and Sophie Schmidt for the illustration of Fig. 7. This work is a contribution of the project "Reduced Complexity Models" (supported by the Helmholtz Association of German Research Centres (HGF) – grant no. ZT-I-0010).





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




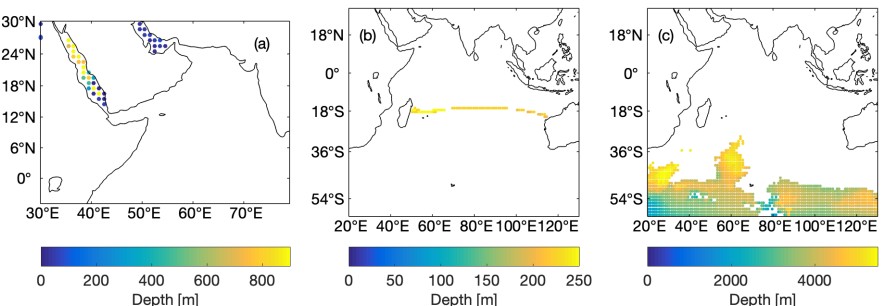

**Figure 1.** Origins of the water mass formation regions from observations (WOA13) for a) Red Sea and Persian Gulf Water, b) Indian Central Water and c) Indian Ocean Deep Water. The colors indicate the deepest depth at each grid point, where the respective water mass properties are found.





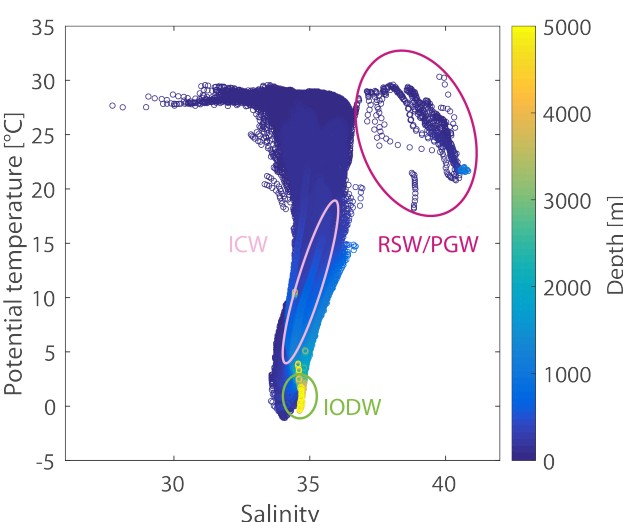

**Figure 2.** TS diagram of the Indian Ocean from observational data (WOA13) color coded by depth. The source water masses for the water mass mixing analysis are Indian Ocean Deep Water (IODW), Indian Central Water (ICW) and Red Sea and Persian Gulf Water (RSW/PGW).




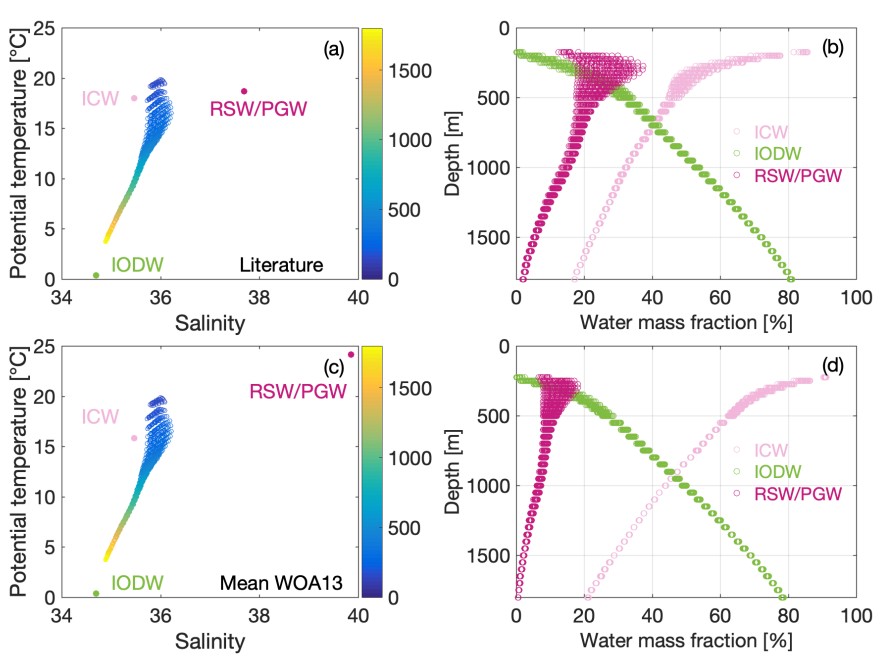

**Figure 3.** TS diagram of the Arabian Sea OMZ with the source water mass properties for the water mass mixing analysis (Indian Ocean Deep Water (IODW), Indian Central Water (ICW) and Red Sea and Persian Gulf Water (RSW/PGW)) defined from a) literature values and c) the averaged observational data (WOA13) and the resulting water mass mixing fractures (b, d).




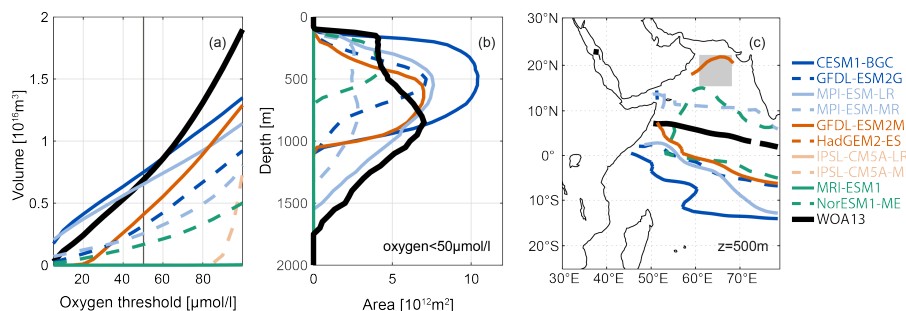

**Figure 4.** Comparison of the Arabian Sea OMZ in observations (WOA13, black) and the CMIP5 models (colored): a) OMZ volume for different oxygen thresholds. The vertical grey line marks the 50 $\mu$mol l$^{-1}$ threshold for the other panels. b) Area of the OMZ for a threshold of 50 $\mu$mol l$^{-1}$ at each depth. c) Map of the OMZ area as defined in b) at 500 m depth. The grey box marks the area of the averaged vertical profiles shown in Fig. 5. Different colors refer to the different model clusters (see text).



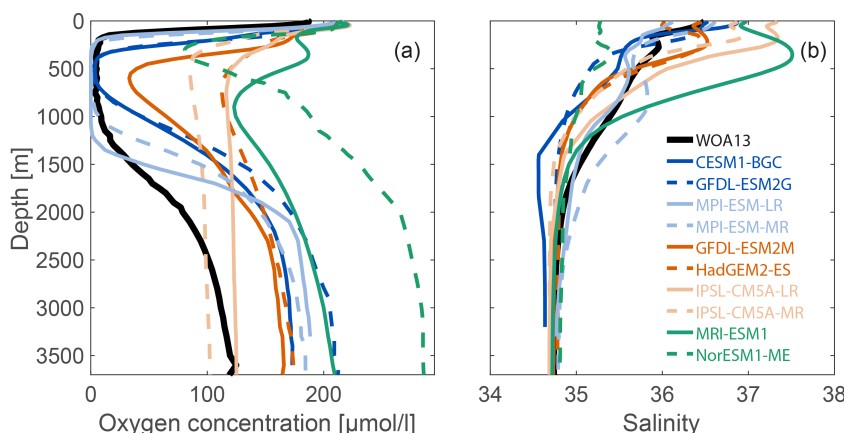

**Figure 5.** Averaged vertical a) oxygen and b) salinity profiles in the box between 16 and 22 °N, 61 and 67 °E (see Fig. 4) in the Arabian Sea for CMIP5 models (colored) and observational data (black). Blue colored models belong to oxygen cluster *HIGH* , red to cluster *MEDIUM* and green to cluster *LOW1* and *LOW2*.





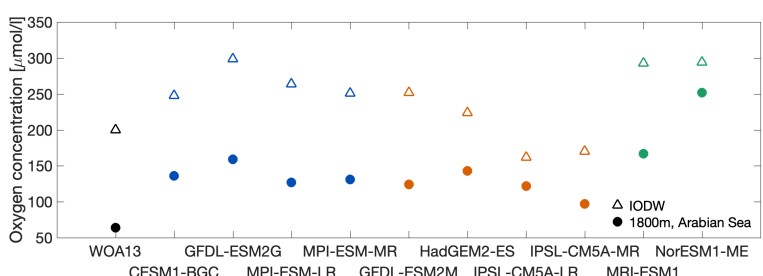

**Figure 6.** Mean oxygen concentration of IODW at its formation site (triangles) and oxygen concentration at the bottom of the OMZ at 1800 m depth in the AS (circles). The colors merk the oxygen clusters as described in Fig. 5.





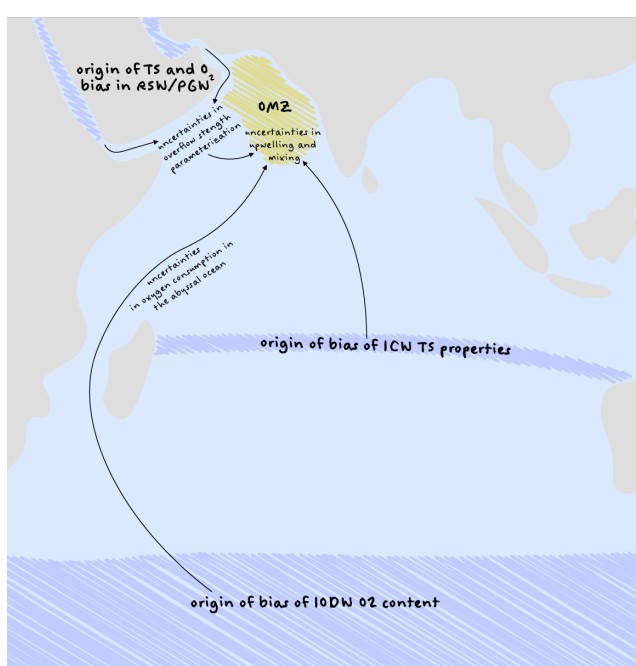

**Figure 7.** Overview sketch of the analysed origins of model-data misfits in oxygen in CMIP5 models. The blue shaded areas mark the origins of the water masses and their related biases in the models. The arrows sketch the way into the OMZ and uncertainties on the way. The yellow shaded area sketches the OMZ in the Arabian Sea.



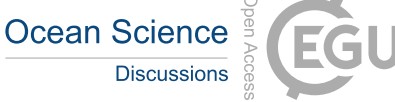
**Table 1.** Summarized information of CMIP5 ocean model components and respective references. IPSL-CM5A-LR and -MR differ only in the atmospheric horizontal resolution, with similar ocean modules in both model setups.

| Model | Resolution (lon/lat; depth) | Circulation model | Reference | Biogeochemical model | Reference |
|---|---|---|---|---|---|
| CESM-BGC | 1.125/0.27-0.53; 60 | CCSM4 | Gent et al. (2011) Danabasoglu et al. (2012) | MET | Moore et al. (2004) |
| GFDL-ESM2G | 1/0.3-1; 63 | GOLD | Dunne et al. (2012) | TOPAZ2 | Dunne et al. (2013) |
| MPI-ESM-LR | 1.5/1.5; 40 | MPIOM | Giorgetta et al. (2013) Jungclaus et al. (2013) | HAMOCC5.2 | Ilyina et al. (2013) |
| MPI-ESM-MR | 0.4/0.4; 40 | MPIOM | Giorgetta et al. (2013) Jungclaus et al. (2013) | HAMOCC5.2 | Ilyina et al. (2013) |
| GFDL-ESM2M | 1/0.3-1; 50 | MOM4.1 | Dunne et al. (2012) | TOPAZ2 | Dunne et al. (2013) |
| HadGEM2-CC | 1/0.3-1; 40 | HadGEM2 | Jones et al. (2011) | Diat-HadOCC | Palmer and Totterdell (2001) Halloran et al. (2010) |
| IPSL-CM5A-LR | 2/0.5-2; 31 | NEMOv3.2 | Dufresne et al. (2013) | PISCES | Aumont and Bopp (2006) Séférian et al. (2013) |
| IPSL-CM5A-MR | 2/0.5-2; 31 | NEMOv3.2 | Dufresne et al. (2013) | PISCES | Aumont and Bopp (2006) Séférian et al. (2013) |
| MRI-ESM1 | 1/0.5; 51 | MRI COM | Adachi et al. (2013) | NPZD | Adachi et al. (2013) |
| NorESM1-ME | 1/1.25; 53 | MICOM | Bentsen et al. (2013) | HAMOCC5.1 | Assmann et al. (2010) |




**Table 2.** Overview of water mass characteristics in observations, literature and the CMIP5 models. Temperature and salinity values are given as upper and lower limits of the water masses. Oxygen is given as mean values. Literature values are taken from Hupe and Karstensen (2000; IODW, RSW/PGW) and Acharya and Panigrahi (2016; ICW).

| | IODW | | | RSW/PGW | | | ICW | | | Cluster | |
| --- | --- | --- | --- | --- | --- | --- | --- | --- | --- | --- | --- |
| | Temp. [°C] | Salinity | $O_2$ [μmol/l] | Temp. [°C] | Salinity | $O_2$ [μmol/l] | Temp. [°C] | Salinity | $O_2$ [μmol/l] | $O_2$ | Salinity |
| WOA | 0.0 - 1.0 | 34.65 - 34.7 | 200 | 18.1 - 29.8 | 37.14 - 40.85 | 128 | 7.7 - 15.8 | 34.57 - 35.57 | 200 | HIGH | HIGH |
| Literature | 0.38 | 34.7 | 221 | 18.7 | 37.69 | 53 | 9.0 - 18.0 | 34.5 - 35.5 | 253 - 274 | | |
| CESM-BGC | 0.3 - 0.9 | 34.58 - 34.62 | 248 | 22.8 - 29.0 | 38.66 - 43.27 | 179 | 9.0 - 17.1 | 34.98 - 36.13 | 170 | HIGH | HIGH |
| GFDL-ESM2G | -0.9 - 0.2 | 34.52 - 34.66 | 299 | 20.0 - 26.3 | 37.00 - 41.07 | 180 | 9.8 - 17.7 | 34.49 - 35.29 | 208 | HIGH | MEDIUM |
| MPI-ESM-LR | 0.0 - 0.8 | 34.58 - 34.65 | 264 | 22.5 - 29.3 | 36.68 - 39.28 | 187 | 11.1 - 19.1 | 34.52 - 35.55 | 214 | HIGH | HIGH |
| MPI-ESM-MR | 0.0 - 1.1 | 34.59 - 34.68 | 251 | 21.8 - 29.4 | 36.90 - 42.45 | 197 | 10.8 - 17.7 | 34.76 - 35.68 | 214 | HIGH | HIGH |
| GFDL-ESM2M | 0.5 - 1.9 | 34.56 - 34.73 | 252 | 22.8 - 28.4 | 38.19 - 41.24 | 180 | 10.3 - 18.1 | 34.58 - 35.30 | 196 | MEDIUM | MEDIUM |
| HadGEM2-CC | 0.0 - 1.0 | 34.69 - 34.76 | 224 | 19.4 - 31.2 | 37.00 - 44.84 | 199 | 12.5 - 17.9 | 34.47 - 35.48 | 233 | MEDIUM | MEDIUM |
| IPSL-CM5A-LR | -0.6 - 0.6 | 34.57 - 34.63 | 162 | 18.5 - 26.6 | 39.16 - 46.55 | 212 | 7.0 - 15.7 | 34.87 - 35.69 | 220 | MEDIUM | MEDIUM |
| IPSL-CM5A-MR | -0.6 - 0.4 | 34.58 - 34.67 | 170 | 20.0 - 28.9 | 38.90 - 46.71 | 201 | 7.1 - 16.2 | 34.72 - 35.53 | 215 | MEDIUM | MEDIUM |
| MRI-ESM1 | 0.7 - 1.7 | 34.62 - 34.71 | 293 | 19.5 - 26.9 | 37.82 - 44.54 | 213 | 9.5 - 18.8 | 34.70 - 35.91 | 214 | LOW1 | LOW1 |
| NorESM1-ME | -0.4 - 1.2 | 34.66 - 34.87 | 294 | 20.5 - 25.3 | 36.07 - 41.18 | 215 | 10.9 - 19.9 | 34.50 - 35.40 | 215 | LOW2 | LOW2 |