# Peer review of "Causes of uncertainties in the representation of the Arabian Sea oxygen minimum zone in CMIP5 models"

_Ocean Science, 2021_

## Referee Comment (RC2)

**Title: Causes of uncertainties in the representation of the Arabian Sea oxygen minimum zone in CMIP5 models**

**General Comments:**

The authors present an interesting attempt to answer some of the uncertainties in the OMZ representation using the 10 CMIP5 model outputs. This is an important issue as the future prediction of OMZ shows a large inter-model spread which is a reflection of the current status of model OMZs. The models are first clustered into four bands based on vertical profiles of oxygen and then the models exhibiting most similarity with are observations are compared with other models in their water mass formation process — a representative of ventilation. The authors primarily attribute the uncertainties to a higher oxygen content in the southern ocean and the coarser model vertical resolution. However, quantitative assessment of the model discrepancies is not possible by just looking at model outputs, but I understand this is beyond the scope of the study.

The authors have communicated the scientific evidences with good clarity — outlining the scientific methods and results clearly — which is appreciable. The discussion section can be written a bit more orderly. Scientifically, there is very little discussion of the respiration part other than in the introduction section. Also, the methodology part could be a bit more explanatory to ensure transparency. Overall, the manuscript discusses a very important problem seen in nearly all the earth system models and is promising. I therefore, recommend the study alongside stating my serious concerns in detail below.

**Specific Comments:**

1. One cause of deoxygenation is solubility which is highly sensitive to the temperature of the oceans — warmer the surface ocean, lesser the solubility of oxygen into the ocean. The representation of ASOMZ in the CMIP5 models will also be a function of ocean temperature in the surface ocean waters. The authors discuss ventilation and respiration to some extent — which I agree are the major causes of deoxygenation. However, I feel the solubility factor might also account to some of the OMZ difference seen among the models. How would the authors justify not looking at the solubility parameter while accounting for the uncertainty in CMIP5 models.

2. Discussion**,** Ln 15-24: The authors conclude that in their study there is no definite linkage found between the model resolution and the representation of OMZ — which to me is surprising. Fundamentally, the ASOMZ is located in what we call the shadow zone where ventilation occurs through mixing processes mainly caused by mesoscale eddies [Resplandy et al., 2012, Lachkar et al., 2016]. Increasing the model horizontal resolution should result in more mesoscale eddy activity hence allowing more ventilation (due to eddy mixing) and hence changing the OMZ. An absence of a linkage between the OMZ and model resolution highlights serious issues in the parameterization of subgrid scale processes in the models or the possibility that the increased model ventilation is being balanced by an increase in model respiration. I am not sure if you can conclude that increasing the horizontal resolution has no effect on the OMZ. Please explain your disagreement to my explanation, if any, and state your reasons for making such a conclusion.

3. Statement: "There is some evidence that these model flaws are related to a deficient representation of ventilation pathways in models. On this basis, it is hardly possible to say whether the models' biogeochemistry does have deficiencies that are associated with the oxygen representation"

   Comment: OMZ is shaped majorly by respiration and ventilation. The study highlights the difference in model mixing of different water masses in the OMZ. However, arising from the fact that OMZ are located in the world's major upwelling zones — which depicts the importance of respiration in shaping OMZ — it is very possible that the model's biogeochemical component is highly (if not equally) responsible for the OMZ volume simulated in the models. Previous studies have addressed the weak representation of biophysical processes in the model, which lays a strong possibility of the deficiencies in the biogeochemical component thus, shaping the model's OMZ. Most of the burden here is placed on the physical parameterizations, whereas the biogeochemical part is a little under looked. Please justify.

4. If you notice the core region of ASOMZ, then you would find almost all the models largely overestimating primary production in the region. This to me, suggests that large respiration should be occurring in the models. I suggest to check whether respiration is well represented in the models. This will confirm that the problem lies largely with the physical processes or biological processes.

5. What clustering method is performed to identify the clusters. Please add some details about the clustering technique in the methods section of the manuscript.

6. There are 10 CMIP5 models used in the study. However, there are ~15-16 ESM models which participated in CMIP5. Please state your choice to choose these 10 models and leave the others.

7. What is the reason to choose the oxygen threshold value of 60 mircomol/litre?

8. Use of T-S diagrams to resolve the water mass characteristics are not quite effective near the shelves. In such a case, how reliable are the estimates taken for the RSW and PGW water masses. Can this be the reason for the models showing large deficiencies in the RSW and PGW masses. What is the author's opinion?

9. Figure 4a: The authors have shown a vertical line of 50 micromol/litre. Why is this value pointed out when the threshold for hypoxia is considered as 60 in the rest of the study.

10. It is advised to shorten the discussion section slightly. Nonetheless, it can be organized a bit more clearly.

11. Summary, Ln 14-15: The authors suggest improved parameterizations of Persian gulf and Red sea water masses in the models. However, I am not confident if improvement in the parameterization of these water mass overflows into the OMZ region would significantly improve the OMZ. Instead, a more local process improvement would be inclusion of eddies into the parameterizations which would affect the ventilation. Please provide strong evidence supporting your solution put forward to the problem under question.

**12.** The authors discuss a very important problem using the available model outputs. However, using the model outputs it is very difficult to quantitatively separate the discrepancies in the physical and biogeochemical processes. It would be interesting to have a quantitative estimation of the model discrepancies in between the individual processes using some model experiments. I understand this is beyond the scope of this paper, but this can be mentioned as a future scope of the work undertaken.

**13.** The authors start the discussion section mentioning that all the models underestimate ASOMZ [as seen in Fig4a]. However, if we look at the vertical profile of oxygen in core region of OMZ [fig 5] we see that almost all the models have higher concentrations of oxygen. Is this possible that the models are overestimating oxygen in core region of OMZ? Please clarify.

**14.** It is advised to include how the water masses are calculated in the methods section.

**15. Technical Corrections:**

1. Summary, Ln 10: "overestimate oxygen concentrations…." —> I think it should be underestimate.
2. Introduction, Ln 16-17: Provide references.
3. Discussion, Ln 16: "Recent studies analyzing…." —> Provide the references.
4. It is advised to rephrase a few sentences in the discussion section as they are confusing.
5. Summary: Rephrase the first two sentences

---

## Author Response (AR1)

The point-by-point response to the reviews includes all comments from the three anonymous referees. The answers, however, may differ slightly from the original answers that have been posted in the public discussion as they have been updated and adjusted to the revised manuscript. The page and line specifications refer to the revised version of the manuscript.

**Anonymous Referee #1**

This manuscript provides valuable insights into the misrepresentation of the Arabian Oxygen minimum zone (ASOMZ) in 10 historical CMIP5 model simulations, and relates these model deficiencies to the analysis of the different water masses that ventilate this OMZ. Overall, I have found the paper very useful in providing metrics to quantify the representation of the OMZ in these models. The approach of relating deficiencies at different depth of the water column to water masses of different origins provides a new and original understanding of the ventilation pathways of this OMZ, that complements nicely a previous study by the same first author in 2020. The main finding is that CMIP5 models tend to underestimate the lower part of the OMZ due to ventilation of highly oxygenated waters from the Southern Ocean.
I am confident that this work should eventually provide a very valuable contribution but I have a major concern, which is that I found that the writing was often clumsy, sometimes to a point where I was not sure I understood the meaning correctly. Although I did find the ideas and general approach of the paper very promising, reading it was not as pleasant as it could have been and I had to struggle my way through.

> **Reply to reviewer #1**
> **We would like to thank the reviewer for taking the time and for providing feedback to improve the manuscript.**

My major problem was that I could not really understand how water masses were determined based on my reading of 3.3. For example, I did not understand how the formation regions have been localized (lines 5-6-7). I also did not understand how the water mass properties (T-S) were derived from observations. Therefore, it was difficult to follow 4.3 (water representation in models).

> **We thank you for raising the point, that the determination and localization of the water masses in the models is hard to understand. We will rewrite that part for better understanding:**
>
> **p. 6, l. 12 ff**
> **Red Sea Water and Persian Gulf Water (RSW/PGW) are geographically restricted in their formation regions. Figure 2a shows the formation region for RSW/PGW for which temperature and salinity ranges and mean values are determined (Tab. 2 & Fig. S4). In contrast Indian Central Water (ICW) is not geographically restricted in its formation regions. ICW is a mixed water mass and is characterised by a nearly linear temperature-salinity relation that is density-compensated (Tomczak, 1984) and can be identified in T-S diagrams. With this relation, we were able to define upper and lower temperature and salinity limits of ICW in observations and compared those to respective values from the literature (see Tab. 2; Acharya and Panigrahi, 2016). ICW is formed**

**along zonally oriented fronts in the tropical ocean sub-surface layers (Tomczak, 1984). Sprintall and Tomczak (1993) and Schott and McCreary (2001) described the geographical location of the formation region of ICW. Figure 2b shows the grid boxes where these T-S properties are found in the IO in WOA13 observations. These are in line with the description of the formation region as shown by Sprintall and Tomczak (1993) and Schott and McCreary (2001). To investigate the formation region of ICW in the models, we followed the same procedure as previously described for the observations. The linear temperature-salinity relation as given by the T-S diagrams of the individual models (Fig. S5) sets the upper and lower temperature and salinity limits (see also Tab. 2 & Fig. S4c). In contrast to the observations and the literature, the resulting locations that determine the formation region of the simulated ICW are not restricted to the subduction area of ICW. For consistency we limit the formation region of ICW in the models to the subduction area of ICW as prescribed according to Sprintall and Tomczak (1993) andSchott and McCreary (2001): We exclude grid boxes with similar T-S properties that are found outside the subduction region. We also exclude grid boxes within the upper 200 m to analyse the oxygen content of permanently subducted ICW below the mixed layer depth that is transported to the AS and not reventilated into the seasonally varying well ventilated mixed layer. Figure S2 shows the respective area for each model and the deepest depth at each location, where the T-S properties are found. Indian Ocean Deep Water (IODW) originates in the Southern Ocean, where it is often referred to as Circumpolar Deep Water and Antarctic Bottom Water, before it travels northward into the deep IO and mixes along its way with the surrounding water masses. IODW is thus defined as the densest water mass in the IO north of 60 °S that is found below 1500 m depth (Talley et al., 2011a, b). Figure 2c shows the formation region of IODW derived from observations. For this region temperature and salinity limits are determined. IODW in the models is defined in the similar way as in observations. In the models the derived formation regions of IODW in the Southern Ocean differ from those we find in observations (Fig. S3). The oxygen content of the water masses as listed in Tab. 2 and shown in Fig. S4 is calculated, for each model and the observations, by the arithmetic mean of all grid boxes of the corresponding source waters.**

Major understanding issues also involved how Figure 1 was generated, how IODW, ICW, RSW/PGW were identified from Figure 2 (where do the ovals come from?).

**The revision of section 3.3 also includes a more detailed description of the generation of Figure 1 (now Fig. 2). The ovals in Figure 2 (now Fig. 3) sketch the limits of the source water mass properties. We now include that in the caption:**

**TS diagram of the Indian Ocean from observational data (WOA13) color coded by depth. The source water masses for the water mass mixing analysis are Indian Ocean DeepWater (IODW), Indian CentralWater (ICW) and Red Sea and Persian GulfWater (RSW/PGW). The ovals indicate the approximate TS ranges of the respective water masses. Exact values of the water mass properties used in this study can be taken from Fig. 4 and Tab. 2.**

I really liked Figure 4, which is a very nice and synthetic way of representing the OMZ, but I add difficulties because of too many lines on the same plot. I would suggest to have more

panels, for instance to group them by set of clusters instead of showing all models together, with WOA in all of them (which would make 4 clusters x 3 panels= 12 panels).

> **Thank you for raising this point. However, at this stage of the manuscript the clusters have not been introduced. The key massage of that figure is to get a quick overview of all models and to show the overall wide range in the oxygen representation. Thus, we prefer to keep the figure as it is.**

I have the same comment for Figure 5.

> **However, for Figure 5 we really like your suggestion. In the revised manuscript the panels are divided into the individual clusters. This makes it easier for the reader to capture the differences in the individual clusters.**

Also I think it would be easier if the information contained in Table 2 was somehow shown in a set of figures, that would ease the presentation of results and the discussion.

> **Thank you for your suggestion. In addition to Table 2, we will include Figure S4 visualizing the information on the water mass properties given in the Table in the supplement of the revised manuscript.**

Information about the age tracer should also be shown in a synthetic figure.

> **We agree with the reviewer and will add Figure S6 to the supplement visualizing the information about the age tracer that is given in the text.**

In the end, because of my misunderstanding, my review is rather limited in terms of how I am able to evaluate the methodology and conclusions, and I believe that the presentation issues that I've raised must be fixed before a full assement of the content can be provided.

**Anonymous Referee #2**

General Comments:
The authors present an interesting attempt to answer some of the uncertainties in the OMZ representation using the 10 CMIP5 model outputs. This is an important issue as the future prediction of OMZ shows a large inter-model spread which is a reflection of the current status of model OMZs. The models are first clustered into four bands based on vertical profiles of oxygen and then the models exhibiting most similarity with are observations are compared with other models in their water mass formation process — a representative of ventilation. The authors primarily attribute the uncertainties to a higher oxygen content in the southern ocean and the coarser model vertical resolution. However, quantitative assessment of the model discrepancies is not possible by just looking at model outputs, but I understand this is beyond the scope of the study.

The authors have communicated the scientific evidences with good clarity — outlining the scientific methods and results clearly — which is appreciable. The discussion section can be written a bit more orderly. Scientifically, there is very little discussion of the respiration part other than in the introduction section. Also, the methodology part could be a bit more explanatory to ensure transparency. Overall, the manuscript discusses a very important problem seen in nearly all the earth system models and is promising. I therefore, recommend the study alongside stating my serious concerns in detail below.

> **Reply to reviewer #2**
> **We would like to thank the reviewer for taking the time and for providing constructive and very specific comments, which will help to improve the manuscript considerably. We will add more details to the methodologies and restructure the discussion part for more transparency. Also, we have carefully addressed the comments. The point-by-point responses to the specific comments follow below.**

Specific Comments:
1. One cause of deoxygenation is solubility which is highly sensitive to the temperature of the oceans — warmer the surface ocean, lesser the solubility of oxygen into the ocean. The representation of ASOMZ in the CMIP5 models will also be a function of ocean temperature in the surface ocean waters. The authors discuss ventilation and respiration to some extent — which I agree are the major causes of deoxygenation. However, I feel the solubility factor might also account to some of the OMZ difference seen among the models. How would the authors justify not looking at the solubility parameter while accounting for the uncertainty in CMIP5 models.

   > **That is a good point that the reviewer mentions. We had a look at the temperature in the upper layers of the Arabian Sea and find slightly lower temperatures there in the models compared to the observations. We compute oxygen solubilities and analyse corresponding model-data differences and will add these findings to the revised manuscript. The temperature profiles are shown in Figure S8 and the solubilities are listed in Table S1. The discussion point is as follows:**
   >
   > **p. 15, l. 28 ff**

In the cluster analysis, offsets in oxygen concentrations between profiles were not considered (Fig. 5a,c,e,g).We focused rather on the shape of the curves, because we regarded the information content as higher for our purposes. The oxygen overestimation of all the considered models at the surface in the AS can be explained by higher oxygen solubilities at the surface in the models of up to 4.7 % compared to observations (Tab. S1). These higher solubilities are caused by lower-than-observed temperatures in the models at the surface (Fig. S8). With the higher solubilities and the positive oxygen offset at the surface in the models, more oxygen could be mixed into the ASOMZ from above than in the observations. Mixing of oxygen from the surface to the interior ocean is dependent on the stratification in the upper ocean as well as the oxygen gradient. The averaged stratification over the box in the AS in the models strongly resembles the observational stratification (Fig. S9). Furthermore, all models and the observations show a strong oxygen gradient above the ASOMZ. Thus it is possible that a small proportion of the overestimated oxygen concentrations in the models could be explained by solubility differences at the surface of the AS.**

2. Discussion, Ln 15-24: The authors conclude that in their study there is no definite linkage found between the model resolution and the representation of OMZ — which to me is surprising. Fundamentally, the ASOMZ is located in what we call the shadow zone where ventilation occurs through mixing processes mainly caused by mesoscale eddies [Resplandy et al., 2012, Lachkar et al., 2016]. Increasing the model horizontal resolution should result in more mesoscale eddy activity hence allowing more ventilation (due to eddy mixing) and hence changing the OMZ. An absence of a linkage between the OMZ and model resolution highlights serious issues in the parameterization of subgrid scale processes in the models or the possibility that the increased model ventilation is being balanced by an increase in model respiration. I am not sure if you can conclude that increasing the horizontal resolution has no effect on the OMZ. Please explain your disagreement to my explanation, if any, and state your reasons for making such a conclusion.

**Thanks for raising this point. In principle we agree with the reviewer. However, in all models considered in our study the sub grid processes are parameterized. Even those models that have a higher resolution are only eddy permitting. The comparison of these models shows that there is no improvement with resolution of the OMZ representation as long as all models are non eddy resolving. We agree with the reviewer that regional models eddy resolving resolutions definitely improve the representation of the OMZ. We will clarify this point in the discussion to avoid misunderstanding of our conclusions:**

**p. 15, l. 11 ff**
**Among the non-eddy-resolving CMIP5 models considered here, we confirm the lack of an apparent systematic coherence between model resolution and better representation of the ASOMZ (Tab. 1 & 2). This is not what we expect from the results of regional eddy-resolving models, i.e. that ventilation of the ASOMZ occurs through mixing processes mainly related to mesoscale eddies (e.g. Resplandy et al., 2012; Lachkar et al., 2016). An increased horizontal resolution of the model should therefore lead to more explicitly resolved mesoscale eddy activity, which might allow for more ventilation and thus a change in the ASOMZ. It seems that resolving mesoscale eddies leads to substantial**

**improvements in the representation of the ASOMZ (Resplandy et al., 2012; Lachkar et al., 2016). However, moving from the range of non-eddy-resolving models to eddy-permitting models, a higher resolution seems to have a minor effect on the ASOMZ.**

3. Statement: "There is some evidence that these model flaws are related to a deficient representation of ventilation pathways in models. On this basis, it is hardly possible to say whether the models' biogeochemistry does have deficiencies that are associated with the oxygen representation"
Comment: OMZ is shaped majorly by respiration and ventilation. The study highlights the difference in model mixing of different water masses in the OMZ. However, arising from the fact that OMZ are located in the world's major upwelling zones — which depicts the importance of respiration in shaping OMZ — it is very possible that the model's biogeochemical component is highly (if not equally) responsible for the OMZ volume simulated in the models. Previous studies have addressed the weak representation of biophysical processes in the model, which lays a strong possibility of the deficiencies in the biogeochemical component thus, shaping the model's OMZ. Most of the burden here is placed on the physical parameterizations, whereas the biogeochemical part is a little under looked. Please justify.

> **You are right. The OMZ volume simulated in the models depends strongly on the models' biogeochemistry as well as the representation of circulation.**
> **The focus of this paper, however, is placed on the physical processes. The underlying physical circulation has a large impact on the biogeochemical model components. Deficiencies of the physical circulation model can be compensated by over-tuning the biogeochemical model. This can be illustrated with the given example: If the upwelling strength in the model is deviating from the observations, this would strongly affect the nutrient supply in the AS and thus the phytoplankton growth that influences the respiration.**
> **We have realised that there is an imbalance between the introduction and the discussion in the manuscript related to the consideration of biogeochemical processes and model uncertainties. We will clarify in the introduction that the focus lies on the physical model components.**
>
> **p. 3, l. 29 ff**
> **There is some evidence that modelled thermocline OMZs are particularly sensitive to applied wind forcing (Oschlies et al., 2017) and that these model flaws are related to a deficient representation of ventilation pathways in models. As the underlying physics influence the biogeochemical model components, there is some risk that errors in the physics may be compensated by errors in the biogeochemical model components (Löptien and Dietze, 2019). Therefore, we consider it important and prudent to evaluate the model physics first before addressing possible errors in the model biogeochemistry.Without a proper evaluation of the model physics, it is hardly possible to say whether the models' biogeochemistry does have deficiencies that are associated with the oxygen representation (Oschlies et al., 2018; Segschneider and Bendtsen, 2013).**

4. If you notice the core region of ASOMZ, then you would find almost all the models largely overestimating primary production in the region. This to me, suggests that large respiration should be occurring in the models. I suggest to check whether respiration is

well represented in the models. This will confirm that the problem lies largely with the physical processes or biological processes.

**We do not find any confirmation in the literature that almost all the models largely overestimate primary production in the ASOMZ region. Based on the historical data of the CMIP5 models, Bopp et al. (2013) shows that the multi model mean underestimates the NPP in the AS in the upper 600 m. In detail, Roxy et al. (2016) looked at the chlorophyll bloom. According to them, only three out of the here considered 10 models overestimate the chlorophyll bloom in the AS. Two of these models are from cluster HIGH and one belongs to cluster MEDIUM. Thus, we cannot confirm where the uncertainties come from.**

5. What clustering method is performed to identify the clusters. Please add some details about the clustering technique in the methods section of the manuscript.

    **As written in the methods section 3.2 we used the Hierarchical Agglomerative Cluster Analysis that was introduced by Johnson (1967). With this method we clustered the correlation between the vertical profiles in the Arabian Sea for oxygen and for salinity separately. We will add some more details in the revised manuscript:**

    **p. 5, l. 18 ff**
    **To reduce the large amount of model output data and detect similarities between the models and observations we grouped them with the Hierarchical Agglomerative Cluster Analysis (Johnson, 1967). Here, the correlation between the vertical oxygen profiles was used as the distance measure for the clusters. This means that profiles that are more similar to each other than to others are grouped together in a cluster.**

6. There are 10 CMIP5 models used in the study. However, there are ~15-16 ESM models which participated in CMIP5. Please state your choice to choose these 10 models and leave the others.

    **We chose the 10 models from the CMIP5 models that provided oxygen data for the historical period. We will clarify that in the revised manuscript:**

    **p. 4, l. 18 f**
    **In this study we included all ESMs from the CMIP5 project (Taylor et al., 2012), where output of dissolved oxygen was available. The suit of ten model simulations includes …**

7. What is the reason to choose the oxygen threshold value of 60 mircomol/litre?

    **Unfortunately this is a misunderstanding. We did not choose an oxygen threshold for the analysis but used averaged oxygen profiles in order to be able to compare the OMZs in a way that is as generally valid as possible. The thresholds that are mentioned in the text are used to make different statements, as the behaviour among the models show systematic differences when accounting a specific threshold. We will clarify this in the revised manuscript.**

> **In addition, we add two vertical lines to Figure 4a (now 1a) for more clarity. With this modification, all thesholds that are mentioned in the text are included in the figure.**

8. Use of T-S diagrams to resolve the water mass characteristics are not quite effective near the shelves. In such a case, how reliable are the estimates taken for the RSW and PGW water masses. Can this be the reason for the models showing large deficiencies in the RSW and PGW masses. What is the author's opinion?

> **This is a good point. As the formation regions are clearly defined by the geographical location, the T-S properties given here are exactly as simulated by the models. This does not mean that they are close to the observations. Coastal regions and shelf areas are not well resolved in the coarse resolution models. This might be one reason for the models showing deficiencies in the RSW and PGW water masses. We will include this point to the manuscript:**
>
> **p. 14, l. 12 f**
> **A possible reason for this model-data oxygen difference in RSW/PGW could be the poor resolution of coastal regions and shelf areas in the coarse resolution models, which includes the shallow marginal seas.**

9. Figure 4a: The authors have shown a vertical line of 50 micromol/litre. Why is this value pointed out when the threshold for hypoxia is considered as 60 in the rest of the study.

> **As mentioned in point 7 above, our analysis does not rely on a single threshold for oxygen. To avoid this misunderstanding in the revised manuscript, we clarified this in the text, as well as we modified the figure. For details please see Point 7.**

10. It is advised to shorten the discussion section slightly. Nonetheless, it can be organized a bit more clearly.

> **Thank you for your advice, we will revise the discussion section.**

11. Summary, Ln 14-15: The authors suggest improved parameterizations of Persian gulf and Red sea water masses in the models. However, I am not confident if improvement in the parameterization of these water mass overflows into the OMZ region would significantly improve the OMZ. Instead, a more local process improvement would be inclusion of eddies into the parameterizations which would affect the ventilation. Please provide strong evidence supporting your solution put forward to the problem under question.

> **We agree with the reviewer that eddies might have a large impact on the OMZ. This important aspect will be added to the revised manuscript. However, upon the mismatches we find are in addition deficiencies in the representation of the RSW and PGW. To give a complete picture of potential error sources this needs to be mentioned as well.**
>
> **p. 17, l. 9 ff**
> **We consider it useful to first address local processes that can be clearly delimited and whose uncertainties are not amplified by other errors. These are the parametrisation of the overflow of RSW/PGW and their T-S properties in the**

**source region as well as the better representation of sub-grid scale processes in the AS itself.**

12. The authors discuss a very important problem using the available model outputs. However, using the model outputs it is very difficult to quantitatively separate the discrepancies in the physical and biogeochemical processes. It would be interesting to have a quantitative estimation of the model discrepancies in between the individual processes using some model experiments. I understand this is beyond the scope of this paper, but this can be mentioned as a future scope of the work undertaken.

**Thank you for mentioning this point. This is indeed a future scope and we will mention it in the revised manuscript:**

**p. 16, l. 19 ff**
**Therefore, an important next step would be a quantitative estimate of the model discrepancies between the individual physical and biogeochemical processes that form the ASOMZ (i.e. ventilation time of the OMZ and oxygen consumption within the OMZ).**

13. The authors start the discussion section mentioning that all the models underestimate ASOMZ [as seen in Fig4a]. However, if we look at the vertical profile of oxygen in core region of OMZ [fig 5] we see that almost all the models have higher concentrations of oxygen. Is this possible that the models are overestimating oxygen in core region of OMZ? Please clarify.

**You are absolutely right. Underestimating the oxygen minimum zone and higher than observed oxygen concentrations in the core region of the OMZ do not contradict each other. Looking at the OMZ volume or expansion as shown in Fig. 4 (now Fig. 1) always needs a predefined oxygen threshold to define the OMZ. An underestimated OMZ thus means that the volume of water containing less oxygen than the threshold is smaller than in observations. Therefore, averaged profiles (Fig. 5) show higher oxygen values.**

14. It is advised to include how the water masses are calculated in the methods section.

**We apologise that the description of the water mass calculation was not clear. We will rewrite and rephrase section 3.3 'Determination of water masses in models' to make our method comprehensible:**

**p. 6, l. 12 ff**
**Red Sea Water and Persian Gulf Water (RSW/PGW) are geographically restricted in their formation regions. Figure 2a shows the formation region for RSW/PGW for which temperature and salinity ranges and mean values are determined (Tab. 2 & Fig. S4). In contrast Indian Central Water (ICW) is not geographically restricted in its formation regions. ICW is a mixed water mass and is characterised by a nearly linear temperature-salinity relation that is density-compensated (Tomczak, 1984) and can be identified in T-S diagrams. With this relation, we were able to define upper and lower temperature and salinity limits of ICW in observations and compared those to respective values from the literature (see Tab. 2; Acharya and Panigrahi, 2016). ICW is formed along zonally oriented fronts in the tropical ocean sub-surface layers (Tomczak,**

**1984). Sprintall and Tomczak (1993) and Schott and McCreary (2001) described the geographical location of the formation region of ICW. Figure 2b shows the grid boxes where these T-S properties are found in the IO in WOA13 observations. These are in line with the description of the formation region as shown by Sprintall and Tomczak (1993) and Schott and McCreary (2001). To investigate the formation region of ICW in the models, we followed the same procedure as previously described for the observations. The linear temperature-salinity relation as given by the T-S diagrams of the individual models (Fig. S5) sets the upper and lower temperature and salinity limits (see also Tab. 2 & Fig. S4c). In contrast to the observations and the literature, the resulting locations that determine the formation region of the simulated ICW are not restricted to the subduction area of ICW. For consistency we limit the formation region of ICW in the models to the subduction area of ICW as prescribed according to Sprintall and Tomczak (1993) andSchott and McCreary (2001): We exclude grid boxes with similar T-S properties that are found outside the subduction region. We also exclude grid boxes within the upper 200 m to analyse the oxygen content of permanently subducted ICW below the mixed layer depth that is transported to the AS and not reventilated into the seasonally varying well ventilated mixed layer. Figure S2 shows the respective area for each model and the deepest depth at each location, where the T-S properties are found. Indian Ocean Deep Water (IODW) originates in the Southern Ocean, where it is often referred to as Circumpolar Deep Water and Antarctic Bottom Water, before it travels northward into the deep IO and mixes along its way with the surrounding water masses. IODW is thus defined as the densest water mass in the IO north of 60 °S that is found below 1500 m depth (Talley et al., 2011a, b). Figure 2c shows the formation region of IODW derived from observations. For this region temperature and salinity limits are determined. IODW in the models is defined in the similar way as in observations. In the models the derived formation regions of IODW in the Southern Ocean differ from those we find in observations (Fig. S3). The oxygen content of the water masses as listed in Tab. 2 and shown in Fig. S4 is calculated, for each model and the observations, by the arithmetic mean of all grid boxes of the corresponding source waters.**

15. Technical Corrections:

    1. Summary, Ln 10: "overestimate oxygen concentrations…." —> I think it should be underestimate.

**Overestimate is right in this sentence. We say that the models overestimate the overall oxygen concentration in the Arabian Sea, which is right for our study (Fig. 5) as well as for the CMIP6 models (Seferian et al., 2020).**

    2. Introduction, Ln 16-17: Provide references.

**We will add the reference: 'The strong influence of the semi-annually changing monsoon winds on the circulation and resulting upwelling and subduction in the AS shapes the OMZ (Schott & McCreary, 2001; Schmidt et al., 2020).**

    3. Discussion, Ln 16: "Recent studies analyzing…." —> Provide the references.

**We will add the reference: 'Recent studies by Seferian et al. (2020) and Kwiatkowski et al. (2020) analysing CMIP5 and CMIP6 model data show that**

**increasing the horizontal resolution does not overcome the major problems with respect to simulating oxygen in the open ocean.'**

4. It is advised to rephrase a few sentences in the discussion section as they are confusing.

**We will go through the discussion section again and rephrase sentences that are hard to understand.**

5. Summary: Rephrase the first two sentences

**We will rephrase the sentences: 'In this paper we compared 10 ESMs from the CMIP5 historical experiment and analysed their representations of the modelled ASOMZs. We systematically grouped the models with a cluster analysis. By comparing the representation of water masses and mixing in the models with observations, we identified systematic weaknesses in the ESMs that lead to deficient oxygen concentrations in the AS in the northern IO.'**

**Anonymous Referee #3**

General comments:
This paper aims to assess the representation of the Arabian Oxygen minimum zone (ASOMZ) in 10 CMIP5 model historical simulations and relates the error to water mass properties. The topic is interesting and important. However, there are several major issues that should be addressed. The authors stated that none of the selected CMIP5 ESMs reproduces the observed oxygen distribution. It would be interesting to examine these aspects in their upgraded CMIP6 versions to check if they had substantially improved or worsen in representing OMZ and water mass properties.

The water mass properties over the north Indian Ocean region and their implications on ocean biogeochemistry in CMIP models have not been studied extensively. This paper presents important and fresh perspectives through the use clustering and quantification of uncertainties in water mass mixing ratios. However, the paper has not been written clearly. This manuscript was not structured well, especially the introduction and results sections. Introduction needs to be organised. Re-structuring of the manuscript can be done to make it easily readable and highlight the novelty of the study. I would recommend a major revision.

> **Reply to reviewer #3**
> **We would like to thank the reviewer for taking the time and for providing constructive and very specific comments, which will help to improve the manuscript considerably.**
> **We agree with the reviewer that it would be interesting to examine these aspects also for the upgraded CMIP6 models. However, we do not find it advisable to include the analysis of the CMIP6 models in this study. First of all, for a meaningful discussion of our results the previous work done on CMIP5 models by other scientists was important. This enables us to set our results into perspective and to draw conclusions. However, for the relatively new set of CMIP6 models, such a base is so far not available. Second, the protocols for the new CMIP6 models differ from the older CMIP5 ones. With this, they form a new and independent set of experiments and cannot be treated identically to CMIP5. Therefore, a comprehensive discussion and interpretation of the results is only possible to a very limited extent. However, we are aware of the importance to fully investigate the CMIP6 models, therefore we included all relevant available information (and the respective references) in the discussion.**
> **We will revise the introduction and make sure that the novelty of the study is clear throughout the manuscript.**
> **We have carefully addressed all the comments. The point-by-point responses to the specific comments follow below.**

Specific comments:
1. Why did the authors choose 50 threshold to define OMZ? Please clarify in the methodology section.

> **Unfortunately, this is a misunderstanding. We did not choose an oxygen threshold for the analysis but used averaged oxygen profiles in order to be able to compare the OMZs in a way that is as generally valid as possible. The thresholds that are mentioned in the text are used to make different statements, as the**

**behaviour among the models show systematic differences when accounting a specific threshold. We will clarify this in the revised manuscript:**
**In the methods section page 5, line 13 ff we explained the choice of this threshold for the plot. To prevent misunderstandings, we will rewrite the sentence: 'For a first spatial comparison, we chose our threshold to be 50 μmol l−1 to make it comparable to previous studies on CMIP5 oxygen distribution (e.g. Cabré et al., 2015; Cocco et al., 2013) and look at the horizontal extent of the ASOMZ as a function of depth and the actual location of these areas in a map.'**
**In addition, we add two vertical lines to Figure 4a (now Fig. 1) for more clarity. With this modification, all thesholds that are mentioned in the text are included in the figure.**

2. Are there any criteria adopted in selecting the specific ESMs? Are they good at representing the Arabian Sea mean state? Provide references if available.

   **No, there are no criteria for the selection of the models. We chose the 10 models from the CMIP5 models that provided oxygen data for the historical period. We will clarify that in the revised manuscript:**

   **p. 4, l. 18 f**
   **In this study we included all ESMs from the CMIP5 project (Taylor et al., 2012), where output of dissolved oxygen was available. The suit of ten model simulations includes …**

   **As we focus on oxygen, we give an overview of the oxygen mean state in these models (Fig. 4) and see that they are not that good in representing it. Other variables and processes connected to the representation of the OMZ in the Arabian Sea that were already analysed for the CMIP5 models were referenced in the discussion.**

3. Description of OMZ along west coast of India can be included in the introduction section.

   **Thank you for your suggestion. We are aware of the coastal OMZ and the complex dynamics right off the west coast of India. However, the resolution of the ESMs used in this study is too coarse, so coastal processes are not fully resolved and the model bias in these areas is expected to be large. We therefore excluded the coastal areas for the determination of the clusters and focus on the open ocean OMZ.**
   **We will briefly discuss this point in the manuscript and emphasis the central Arbian Sea as the focus area of this study:**

   **p. 6, l. 4 ff**
   **For this analysis we chose to exclude coastal areas, because the model bias in these areas is expected to be large due to the coarse resolution of the ESMs. We focus on the open ocean core of the ASOMZ in the central AS between 16 and 22 °N, 61 and 67 °E and from 10 to 1800 m depth and analysed averaged profiles in this region, which is marked in Figure 1c.**

4. The description of mixing ratio coefficients is not clear. Please elaborate. Define in terms of their corresponding water mass.

**We will specify the description of the mixing ratio coefficient in the revised manuscript, and we will explicitly mention the corresponding water masses used in this context:**

**p. 7, l. 8 ff**
**The three main source water masses in the AS are IODW, RSW/PGW and ICW (Fig. 3).We used a linear mixing approach and restricted the input to physical water mass properties from observational data. By considering potential temperature (θ), salinity (S) and mass conservation this yielded the possibility to resolve the mixing ratio of the three main source water masses in the AS. The set of linear equations was:**
$$\theta = \alpha\theta(IODW) + \beta\theta(ICW) + \gamma\theta(RSW/PGW) \quad (1)$$
$$S = \alpha S(IODW) + \beta S(ICW) + \gamma S(RSW/PGW) \quad (2)$$
$$1 = \alpha + \beta + \gamma \quad (3)$$
**α, β and γ were the mixing ratio coefficients for IODW, ICW and RSW/PGW, respectively.**

5. Apart from the errors associated with ventilation, it would be interesting to describe the static stability and solubility parameter in these models. Stratification of upper layers associated with warming and weakened surface winds restrict mixing oxygen-rich surface waters to intermediate depths, leading to oxygen depletion. Please clarify.

   **That is a good point that was mentioned as well by reviewer #2. We will compute oxygen solubilities and analyze corresponding model-data differences and will add these findings to the revised manuscript. We will also compare the static stability in the upper layers and discuss the findings in the revised manuscript. The results are presented in Figures S8, S9, and Table S1. The discussion point is as follows:**

   **p. 15, l. 28 ff**
   **In the cluster analysis, offsets in oxygen concentrations between profiles were not considered (Fig. 5a,c,e,g).We focused rather on the shape of the curves, because we regarded the information content as higher for our purposes. The oxygen overestimation of all the considered models at the surface in the AS can be explained by higher oxygen solubilities at the surface in the models of up to 4.7 % compared to observations (Tab. S1). These higher solubilities are caused by lower-than-observed temperatures in the models at the surface (Fig. S8). With the higher solubilities and the positive oxygen offset at the surface in the models, more oxygen could be mixed into the ASOMZ from above than in the observations. Mixing of oxygen from the surface to the interior ocean is dependent on the stratification in the upper ocean as well as the oxygen gradient. The averaged stratification over the box in the AS in the models strongly resembles the observational stratification (Fig. S9). Furthermore, all models and the observations show a strong oxygen gradient above the ASOMZ. Thus it is possible that a small proportion of the overestimated oxygen concentrations in the models could be explained by solubility differences at the surface of the AS.**

6. Page 5, line 10: "We chose our threshold to be 50 ". But a threshold of 60 is referred to state the general underestimation of OMZ volume (e.g.: Abstract section). Please clarify.

As explained above (see point 1), we did not choose a single oxygen threshold for the analysis. In the discussion we state that "All ten models underestimate the ASOMZ volume when we consider oxygen thresholds of 60 μmol l−1 or higher (Fig. 4a)." This is just the threshold that fits the statement and is not related to the Figures 4b & c.

To avoid further misunderstanding, we also added the two thresholds of 20 and 60 μmol l−1 to Fig. 1a.

7. Page 16, line 5: "…….physical model components show no obvious deficiencies in circulation and mixing". The analysis presented in this paper is not sufficient to conclude this. Please clarify.

We would like to apologise, as this sentence was misleading. We wanted to say that the physical models show deficiencies, but that these are not large enough to adequately explain the deviations in oxygen. We will rephrase it in the revised manuscript.

Technical corrections:
Page 4, line 20: "……..OMZ between 200 and 1800m". Provide references.

The concrete depth of the OMZ depends on the oxygen threshold and varies among the models. Thus, there are various depth ranges related to the OMZ. We neglected that fact while writing such a general statement that refers to the observations and the threshold of 50 μmol l−1. We apologize for that and will rewrite this sentence:

p. 4, l. 27 ff
Averaging also neglects the seasonal cycle. The seasonal oxygen cycle is weak in the upper layers of the AS and not noticeable at greater depth (Schmidt et al., 2020). Thus, averaging is a reasonable approach for a uniform process analysis over large parts of the water column.

Page 4, line 25: "…….depth levels ranges from 31 to 63". Please rewrite this sentence. What are the numbers 31 and 63?

The numbers are the numbers of resolved depth levels in the models. We rewrote the sentence to make that clear: 'The horizontal resolution ranges from 2° x 2° to 0.4° x 0.4° and the vertical resolution varies between 31 and 63 resolved depth levels.'

Page 5, line 10: "We thus compare the volume of the OMZ for a wide range of thresholds." Please provide the values.

We included the values and the new sentence is: 'We thus compare the volume of the OMZ for a range of thresholds from 0 to 100 μmol l−1.'

Page 5, line 25: "…….Oxygen profiles in the AS for all models and the observations." All models or selected ESMs?

**With all models we mean all the 10 models considered for this study. We changed the sentence to avoid misunderstandings: 'We performed the cluster analysis for oxygen profiles in the AS for all 10 models considered in this study and the observations.'**

Page 5, line 30: Is that the area shown in Fig. 4c? The location of the central AS can be better shown on a map.

**Yes, that is the area marked in Figure 4c. We will include the information to the text and reference the Figure accordingly: 'For this analysis we chose to exclude coastal areas and focus on the open ocean core of the ASOMZ in the central AS between 16 and 22 °N, 61 and 67 °E and from 10 to 1800 m depth and analysed averaged profiles in this region, which is marked in Figure 1c.'**

Page 6, line 30: "...... three different source water masses". Please mention three source water masses.

**We will mention them here: 'The three main source water masses in the AS are IODW, RSW/PGW and ICW (Fig. 3). We used a linear mixing approach and restricted the input to physical water mass properties from observational data. By considering potential temperature ($\theta$), salinity (S) and mass conservation this yielded the possibility to resolve the mixing ratio of the three main source water masses in the AS.'**

Page 7, line 5:" ......IODW, RSW and PGW and ICW". Please rewrite this sentence. Should it be like.........PGW/RSW? Please provide proper references to the methods described to determine the source water masses (Page 7, line 5-15).

**We apologize for the deficient description how we determined the source water masses. We will restructure and rewrite parts of that section to make our methods more comprehensible (see also point 4):**

**p. 7, l. 16 ff**
**…the mixing ratio coefficients for IODW, ICW and RSW/PGW, respectively. The equations were solved at each data grid point.**
**We first solved the equations for each observational WOA13 data grid point in the box in the ASOMZ (Fig. 4b) by using observation based temperature and salinity values of the source water masses from literature (Table 2, Fig. 4a). Temperature and salinity values of the source water masses from literature differ to those derived from the WOA13 observations. The same applies for the model temperature and salinity values. In addition, the properties of the water in the ASOMZ in the models differ from each other and from those of the observations. To obtain an uncertainty range of the water mass analysis that can be related to a change of the source water mass input, we solved the equations again for each observational WOA13 data grid point in the box in the ASOMZ, but this time we used arithmetic temperature and salinity mean values of the WOA13 data in the IO, following the calculations described in section 3.3 for oxygen (Fig. 4c & d). This information about the sensitivity of mixing ratios to the definition of water mass properties allows us to draw conclusions on the significance of differences**

**between modelled and observed mixing ratios. Note, the prescribed temperature and salinity values from the source water masses determine the vertical extent of the mixing results and limit our analysis to the central AS and thus the core region of the ASOMZ, which is of the main interest of this study (Fig. 4b & d).**

Provide references or describe the method to obtain the age of water masses in selected models (Page 9, line 30).

**We will include the method how we obtained the age of the water masses in the models:**

**p. 10, l. 12 ff**
**Differences in the transit time can be determined by an age tracer. Only two out of ten models include an ideal age, that is an idealised tracer that counts the time since the last surface contact. We obtained the ideal age of IODW in the Southern Ocean by the arithmetic mean of all grid boxes of the formation region of the source water mass, similar to the calculation of the oxygen content (section 3.3). In the deep AS the ideal age is calculated by the mean within the averaging box of the profiles (Fig. 5) below 1800 m depth.**